# Judge a Book by Its Cover: Investigating Multi-Modal LLMs for Multi-Page Handwritten Document Transcription

## Abstract

Handwriting text recognition (HTR) remains a challenging task. Existing approaches require fine-tuning on labeled data, which is impractical to obtain for real-world problems, or rely on zero-shot tools such as OCR engines and multi-modal LLMs (MLLMs). MLLMs have shown promise both as end-to-end transcribers and as OCR post-processors, but to date there is little empirical research evaluating different MLLM prompting strategies for HTR, particularly for the case of *multi-page documents*. Most handwritten documents are multi-page, and share context such as semantic content and handwriting style across pages, yet MLLMs are typically used for transcription at the page level, meaning they throw away this shared context. They are also typically used as either as text-only post-processors or image-only OCR alternatives, rather than leveraging multiple modes. This paper investigates a suite of methods combining OCR, LLM post-processing and MLLM end-to-end transcription, for the task of zero-shot multi-page handwritten document transcription. We introduce a benchmark for this task from existing single-page datasets, including a new dataset, `Malvern-Hills`. Finally, we introduce OCR+PAGE1 and OCR+PAGEN, prompting strategies for multi-page transcription that outperform existing methods by sharing content across pages while minimizing prompt complexity.

## 1 Introduction

A significant proportion of human written material exists only as physical, handwritten documents. Accurate and cost-effective digitization of such documents would benefit many fields by improving information accessibility, searchability, and ease of processing. Digitized handwritten text could also provide a largely-untapped source of training data for future language models.

While modern optical character recognition (OCR) software is now adept at transcribing machine-printed text, handwritten text recognition (HTR) remains challenging; handwritten documents are often multiple pages long, extremely noisy, and can vary enormously in handwriting style and document structure. State-of-the-art HTR models (Li et al., 2023; Fujitake, 2024) typically combine pre-trained vision Transformers (Dosovitskiy, 2020; Liu et al., 2021; Huang et al., 2022; Xu et al., 2020; Kim et al., 2021) and (small) language models (Devlin, 2018; Liu et al., 2019), and rely on fine-tuning with labeled data to perform well. Unfortunately, labeling this training data, i.e. manually transcribing documents, is usually too expensive and time-consuming to be practical in the real world. For this reason, we are interested in models that can be deployed *zero-shot* — without training/fine-tuning or examples.

There are several zero-shot OCR tools for handwriting, such as Google's Vision API, Amazon's Textract, and Transkribus (Kahle et al., 2017; Nockels et al., 2022). These can perform reasonably well, and are fairly cheap, but still frequently return noisy outputs. Furthermore, they operate at, at-most, the page-level. HTR research and benchmarking is primarily focused on the character-, word-, line- or page-level, despite the fact that most real-world handwritten documents are *multi-page*. This means that (after being scanned/photographed) a document is split over multiple images, which are

highly inter-related: handwriting patterns and quirks, structure and formatting, visual artifacts, and, of course, semantic text content. All of this shared context goes unused by the models and tools discussed above, as they process documents one page at a time.

Large language models (LLMs; Floridi & Chiriatti (2020); Achiam et al. (2023); Zhao et al. (2023)) show promise for addressing these challenges. One example of this is using LLMs as a *post-processor* for correcting cheap and noisy OCR output; we discuss this in Section 2 and include experimental results in Section 5. Furthermore, by leveraging the long context capabilities of modern LLMs (Chen et al., 2023; Liu et al., 2023a; Kim et al., 2024; Karpinska et al., 2024), this can even be done all-at-once for multi-page documents, theoretically enabling shared inter-page context to inform corrections.

Even more promising are multi-modal LLMs (MLLMs; Wu et al. (2023)), LLMs that can accept both text and images as prompts. Despite being general models and not trained explicitly for HTR, commercial MLLMs such as GPT-4O are *very* good at end-to-end handwriting transcription, often much better than OCR tools designed for the purpose (see Section 5). The main downsides of end-to-end transcription with MLLMs are (i) the risk of hallucination; an OCR engine may produce a noisy, erroneous signal, but it is less likely than an MLLM to corrupt a signal by introducing text that is 'reasonable' but incorrect, and therefore more difficult to identify; and (ii) expense; the best commercial LLMs can be expensive at scale — images consume many more tokens than raw text, and transcription is a task that produces a lot of output tokens, which are typically priced higher than input tokens. Using MLLMs as postprocessors rather than end-to-end can partially mitigate these issues, but without access to images they tend to underperform. This trade-off is not well understood, nor have there been significant research efforts to bridge the gap by combining OCR postprocessing and end-to-end vision methods to cost-effectively leverage the benefits of both.

**This paper** aims to address a lack of empirical research into the comparison and combination of OCR tools and MLLMs for HTR, specifically in the zero-shot, multi-page document setting; the setting most frequently faced by practitioners. The contributions of this paper are three-fold:

- We investigate the capabilities of OCR, MLLMs, and combinations of both for *zero-shot, multi-page, handwritten document transcription*, evaluating a suite of methods designed to use text and images, and to use context between pages improve transcription.

- We introduce a new multi-page handwritten document dataset, Malvern-Hills, and synthesize two further multi-page datasets from existing single-page datasets to produce a benchmark for evaluating multi-page transcription.[1]

- We propose OCR+PAGE1 and OCR+PAGEN, simple but effective methods permitting MLLMs to extrapolate information from *a single page image* to improve the transcription accuracy of OCR-generated text, leveraging multi-modality and shared inter-page context to improve transcription accuracy while balancing prompt complexity and token cost.

## 2 RELATED WORK

**Handwriting OCR.** Most OCR engines, including those that can be run locally like Tesseract, are designed for use with printed text and are nearly useless for handwriting. Several commercial OCR engines, such as Google Cloud Vision, Azure AI Vision, Amazon Textract and Transkribus (Kahle et al., 2017; Nockels et al., 2022) are designed for use on handwritten text at the page-level scale.

SOTA HTR and OCR models (Fujitake, 2024; Li et al., 2023; Kim et al., 2021; Huang et al., 2022) are typically based on pre-trained vision Transformers (ViT; Vaswani et al. (2017); Dosovitskiy (2020)) and may include recurrent components like LSTMs or CNNs (Breuel et al., 2013; Azawi et al., 2013; Bora et al., 2020; Yang et al., 2019); the commercial handwriting-capable OCR engines mentioned above are likely similar in architecture to the best of these models, leveraging massive, pre-trained ViTs and language models. In general, such models are only somewhat effective on HTR tasks zero-shot, and SOTA is reached by fine-tuning on labeled data for the specific task. This is fine for benchmarks, but for real-world tasks obtaining labeled training data is often prohibitively expensive. Furthermore, most benchmarks are concerned only with recognition at the character-

---

[1]The datasets and all code for reproducing the experiments in this paper can be found at https://anonymous.4open.science/r/judge-a-book-by-its-cover-70F0.

or line-level. This can, of course, be aggregated to return document-level transcriptions, but this neglects the task of text *detection*, and does not consider incidentals that occur in real documents — headings, figures, scribbles, margin notes, imperfections in image quality, distractors, etc. This paper is concerned with transcription over multi-page documents in a holistic manner.

**LLMs for OCR post-processing.** Several works have investigated improving OCR transcription accuracy with post-processing by a language model (Lund et al., 2011; Schaefer & Neudecker, 2020; Veninga, 2024; Rigaud et al., 2019). LLM-aided OCR is a public tool that uses OCR output with an LLM post-processor to improve OCR transcription accuracy, but the authors do not provide any experimental results demonstrating improvement besides hand-picked examples. Similarly, BetterOCR is a tool that combines results from multiple OCR engines and passes them into an LLM, but only hand-picked examples are provided as experimental results. Furthermore, both tools are only designed for printed text, both operate at the page level (or at the finer-grained 'page chunk' level), and neither process images directly with MLLMs, only using LLMs for post-processing.

Several existing works investigate the use of OCR output, including text and bounding boxes, alongside images as input to MLLMs for document understanding (Wang et al., 2024; Luo et al., 2024; Liao et al., 2025; Wang et al., 2025), but these works also focus on the single page case only, are concerned with tasks like visual question answering rather than transcription, and do not use handwritten data, with the exception of small snippets of handwritten mathematical expressions.

**Benchmarks.** Most OCR benchmarks are for machine-printed text, and only for single pages/images (Liu et al., 2023b), such as receipts (Park et al., 2019; Huang et al., 2019)). Kleister is a pair of multi-page, long-context key entity extraction benchmark tasks, but consists of only machine-printed text (Graliński et al., 2020; Stanisławek et al., 2021).

There are a number of HTR benchmarks, including historical documents, documents not written in English or with Latin characters (Sánchez et al., 2019; Zhang et al., 2019; Causer et al., 2018; Dolfing et al., 2020; Serrano et al., 2010; Wigington et al., 2018; Carbonell et al., 2019; Yu et al., 2021), and transcription of numerical digits or mathematical expressions (Liu et al., 2023b; Yuan et al., 2022; Diem et al., 2014). None are explicitly concerned with multi-page documents, and most are at the line- or word-level.

## 3 OCR+PAGE1 AND OCR+PAGEN: CORRECTION FROM A SINGLE PAGE

We propose two instances of the following prompting strategy for OCR post-processing of multi-page documents: provide the MLLM with the *OCR output for the entire document* as well as *a single page image*. Figures 1a and 1b illustrate the two methods, OCR+PAGE1 and OCR+PAGEN.

OCR+PAGE1 is the simpler strategy: the page image chosen is always the first page of the document. For OCR+PAGEN, a cheap LLM is prompted with the OCR text and asked to *choose* the most promising page image to have access to, which may or may not be the first — it should be clear from the OCR text whether a particular page has more or fewer errors, has a significant amount of reference text or relatively little, etc. OCR+PAGEN adds a small amount of prompt complexity, but can reduce failures caused by unsuitable first pages, such as title pages with limited text. We would expect OCR+PAGEN to perform at least as well as OCR+PAGE1, as OCR+PAGEN can always fall back to OCR+PAGE1 by default.

**Motivation.** OCR+PAGE1 and OCR+PAGEN are based on two assumptions. First, that multi-page documents are likely to be similar in many ways: handwriting is likely consistent for a single document, the semantic content is likely to be highly inter-related, and images of the original documents were likely obtained in a similar fashion (smartphone camera, high-resolution scanner, etc.) so image artifacts are likely similar across pages as well. We believe this is a reasonable assumption, indeed, a 'multi-page document' without at least some shared traits across pages is better characterized not as a multi-page document, but a series of individual documents. The second assumption, which follows from the first, is that OCR errors are likely to be repeated across such similar pages. If a page contains a reasonable amount of text, it likely contains examples of many, if not most, handwriting quirks of the writer; after all, the most common 25 (100) words make up about one third (half) of all written English (Kress & Fry, 2015), there are only 52 letter characters, and character

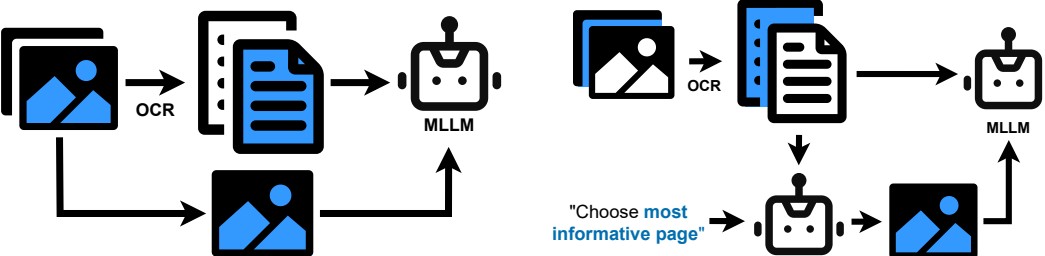

(a) **OCR+PAGE1**: the OCR text of a multi-page document is provided to an MLLM, along with *just the first page image*. Context from the first page is leveraged to correct the OCR output for all pages.

(b) **OCR+PAGEN**: the OCR text of a multi-page document is provided to an MLLM, along with *just the Nth page image*, where $N$ is returned by an upstream LLM with access to the OCR text.

Figure 1: Illustrations of the OCR+PAGE1 and OCR+PAGEN methods. In each case the image passed to the MLLM, and the page corresponding to it in the OCR text, is highlighted. For OCR+PAGEN, in this case, the second page is highlighted, but it could be the first, or any other.

combinations often repeat. Furthermore, challenging words such as proper nouns are reasonably likely to repeat across pages.

We stress that it is not necessary that *all* OCR errors be present in a single page — difficult names may appear once, an individual page may be damaged, etc. — our methods are based on the idea that **a single page provides more useful information than the second**, which provides more useful information than the third, and so on. A lot of page images means redundant information, which means more tokens, more expense and higher prompt complexity.

With these assumptions, we hypothesize that, with OCR+PAGE1 or OCR+PAGEN, an MLLM should be able to *learn, in-context, the mapping from the provided image to the relevant part of the noisy OCR input*, and use this to improve on the post-processing of the entire text. Furthermore, as the OCR transcription for all pages is provided along with an image, semantic content can be shared between pages. This should provide a reasonable trade-off between useful inter-page information and prompt complexity/cost.

A real example from the IAM dataset (see Section 4) of OCR+PAGE1 working in practice is illustrated by Figure 2, and there are several further examples in Appendix B (Figures 8–11).

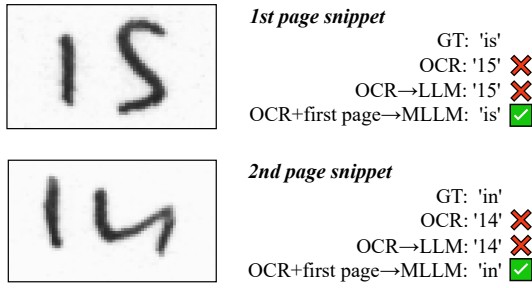

*1st page snippet*

GT: 'is'
OCR: '15' ❌
OCR→LLM: '15' ❌
OCR+first page→MLLM: 'is' ✅

*2nd page snippet*

GT: 'in'
OCR: '14' ❌
OCR→LLM: '14' ❌
OCR+first page→MLLM: 'in' ✅

Figure 2: An example OCR+PAGE1 propagating OCR error corrections across pages. Though the MLLM only has access to the image of the first page, it uses the fact that the OCR (i) frequently mistakes 'i' for '1', and (ii) frequently mistakes words for numbers, to correctly transcribe the word 'in' on the unseen second page.

The methods bears some similarity to few-shot prompting (Dong et al., 2024), in which one or more examples of expected input and desired output are provided within the prompt. The two input modalities for OCR+PAGE1 and OCR+PAGEN, the OCR text and page image, can be thought of as similar to the example input and target. In this case, however, rather than learning in-context to *replicate* the OCR engine's output, the MLLM should (i) exercise its own judgement to identify OCR errors, (ii) identify how the OCR engine's choices should be corrected, and (iii) extrapolate from this learned image and the corresponding OCR text mapping to the remainder of the OCR output (i.e. the 'unseen' text).

## 3.1 A COMPREHENSIVE SUITE OF METHODS FOR EVALUATION

One of the main objectives of this work is to investigate the capabilities of OCR and MLLMs for multi-page transcription by evaluating a comprehensive suite of methods and MLLMs. To our knowledge, no existing works compare and evaluate such a range of methods and prompting strategies for handwriting transcription and OCR post-processing with (M)LLMs. We provide an overview of the methods evaluated in this paper below.

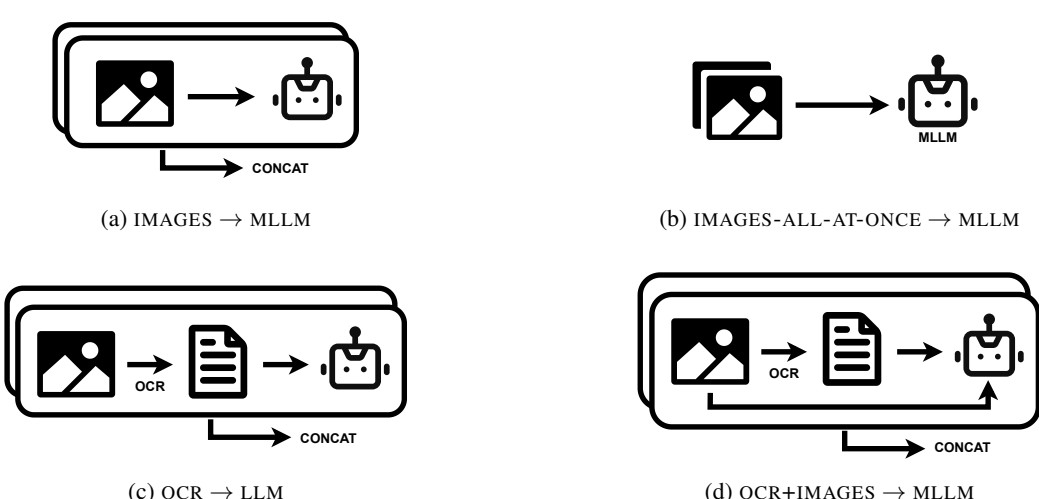

(a) IMAGES → MLLM

(b) IMAGES-ALL-AT-ONCE → MLLM

(c) OCR → LLM

(d) OCR+IMAGES → MLLM

Figure 3: Illustrations of the transcription methods investigated in this paper.

**'All-at-once' and 'page-by-page' processing.** Although we are interested in multi-page documents, it is an essential part of the task that proper page breaks are maintained in the final transcription; for this reason, in our experiments we evaluate transcription accuracy at the page level, rather than the document level, and aggregate. As a result, the methods described below can be classified into 'page-by-page' (PBP), where each page is processed separately, and 'all-at-once' (AAO), where the relevant inputs for all pages are provided, and the MLLM must return JSON-structured output containing a transcription of each page keyed to its given page ID.

For each method below we describe the *input* to the (M)LLM that produces the final transcription. Figures 1 & 3 illustrate these methods to make the pipeline clearer.

**IMAGES** (PBP): the image of a page; equivalent to using an MLLM as an OCR engine end-to-end.

**IMAGES-ALL-AT-ONCE** (AAO): all images in the document; allows access to context across pages, but increases task difficulty and context length

**OCR** (PBP): the output from an OCR engine for a given page; LLM has no access to the original image, but has a smaller token burden, and the potentially easier/more surgical task of *correction* rather than full transcription

**OCR+IMAGES** (PBP): the image of a given page *and* the OCR transcription; we might expect this method to perform the best, as it has the most information, but consume the most tokens

**OCR+PAGE1** (AAO): the image of the *first page* of the document, and JSON input of the OCR transcription of every page (see Figure 1a); our hypothesis is that there is sufficient shared context between pages that a single one will permit effective, token-cheap OCR correction

**OCR+PAGEN** (AAO): the image of a *chosen page* of the document, and JSON input of the OCR transcription of every page (see Figure 1b). The chosen page ID is returned by an upstream (cheap) LLM call, asking, 'given the OCR transcription of the full document, which page image is would a downstream MLLM most benefit from having access to?' We might expect this method to be at least as good as OCR+PAGE1.

## 4 A BENCHMARK FOR MULTI-PAGE HANDWRITTEN DOCUMENT TRANSCRIPTION

In Section 5 we evaluate the methods described above on three multi-page handwritten document datasets, including `Malvern-Hills`, a new dataset labeled by the authors and derived from public domain but not-previously-scanned documents obtained from a charity, the Malvern Hills Trust. See Appendix A for example documents from each dataset.

**IAM.** The IAM Handwriting Database (Marti & Bunke, 2002) is a handwriting benchmark of of single pages, where each page contains a machine-typed passage and a handwritten copy of the same text beneath it, written by one of 657 English-speaking writers. We crop the images to contain only the handwritten part (using provided metadata) and then combine them by writer ID to produce documents with consistent handwriting, and often related content. We use a subset of 242 images to construct 107 multi-page documents, 79 with two pages and 28 with three, and henceforth refer to this multi-page dataset as `IAM`. We note that, though the multi-page documents produced are, in a sense, synthetic, the consistent handwriting means this is suitable for testing the cross-page extrapolation capabilities of our methods. Furthermore, many of the pages in the IAM Database are generated from splitting single text sources anyway (as Figure 5 demonstrates), so many of our multi-page documents do, in fact, flow from page to page and/or share related text content.

**Malvern-Hills.** This dataset is composed of 161 images taken on a smartphone. From these we construct 70 multi-page documents, 49 with two pages and 21 with three. The documents include meeting minutes, correspondence and legal documents, the majority being meeting minutes from between 1889 and 1938. They are written in multiple hands and styles and often use archaic language and handwriting conventions. As a result, this task is more challenging than `IAM`. The documents were initially transcribed by GPT-4O, then manually checked and corrected by the authors. The dataset is included in the linked code repository.

**Bentham.** The Bentham-R0 dataset consists of 433 images of handwritten notes by the 18/19th-century philosopher Jeremy Bentham with crowd-sourced transcriptions (Causer & Wallace, 2012; Causer et al., 2018). Each page is identified by source and page number, but there are many gaps and isolated pages, so we create a multi-page version by extracting all groups of consecutive pages (239 total). The resulting multi-page dataset, `Bentham`, is the most challenging of the three, and consists of 52 two-page documents, 21 three-page documents and 18 four-page documents.

## 5 EXPERIMENTS

Full results are detailed in Tables 1 and 2, with the best models trading off cost against transcription accuracy highlighted by Pareto frontier plots in Figure 4.

### 5.1 EXPERIMENTAL DETAILS

The task for each dataset is to produce transcriptions for each page in the given document, from page images, OCR output derived from those images, or some combination of the two. The methods are described in Section 3.1. All MLLM prompts are included in the linked code repository. Prompts for each method were developed over four rounds of experimentation on a validation split of 100 multi-page documents from IAM Database images separate from the 242 images used to generate `IAM`.

**OCR and MLLMs.** We use the Azure AI Vision OCR engine, as early experimentation comparing Azure AI Vision, Amazon Textract, Google Cloud Vision and Tesseract revealed Azure to be generally the best-performing OCR engine, as well as the cheapest (see Appendix C).

For MLLMs we use two leading commercial models at time of writing, OpenAI's GPT-4O and Google's GEMINI-2.5-PRO, as well as an open-source[2] MLLM, GEMMA-3-27B. For each we use temperature of 0, minimal reasoning (for Gemini) and default parameters otherwise. Details of

---

[2]Under Google's 'open' Gemma license.

cost estimates for OCR and MLLMs are in Appendix D.1. We use a single evaluation run as (i) temperature zero means each output should be near-deterministic, (ii) there is no reason to use a temperature $> 0$ for transcription, as this would mean a chance of sampling output tokens other than those with the highest probability, which would mean a needless error (to put it another way, transcriptions do not benefit from creativity), and (iii) it ensures costs are reasonable for dozens of runs over hundreds of documents.

## 5.2 EVALUATION

Evaluation is primarily at the page level using Character Error Rate (CER; Morris et al. 2004), the most widely-used metric for OCR transcription. We also experimented with Average Normalized Levenshtein Similarity (Peer et al., 2024) and Word Error Rate (WER), but opted for CER as it is more standard, and all three metrics produced very similar relative results.

### 5.2.1 SEMANTIC EVALUATION WITH AN LLM

While CER is a reasonable evaluation metric for transcription, it is based on character-level insertions and deletions and therefore fails to take into account semantic understanding of the document. A transcription may include a correction of a small misspelling present in the original document, use slightly different formatting, spacing or punctuation, or contain other 'errors' that differ from the ground truth labels without affect meaning. Conversely, the incorrect transcription of an initial in a name is only a single character for the purpose of CER, but semantically renders an entire name incorrect. For that reason, we provide an additional semantic evaluation using an LLM on IAM.

For each transcribed page we generate a text diff from the ground-truth using `...<\ins>` and `...<\del>` tags, and pass it to GEMINI-2.5-FLASH with a prompt asking to return a JSON-structured list of all errors, and their classification into one of 7 error types. 'Formatting' and 'semantic' errors, i.e. those which do not affect meaning or remove any information from the original document, are 'minor' errors, whereas other errors: 'missing content', 'hallucination', 'proper noun error', 'numeric error' and 'other' are considered major errors, i.e. errors which impede understanding or remove information. Such an evaluation method is naturally imperfect due to errors in the LLM evaluator, but it is a useful signal, and we found that for a random sample of 5 documents, error classifications were accurate about 95% of the time. Detailed definitions of each error type, and full set of error classifications for IAM can be found in in Appendix C.1.

## 5.3 DISCUSSION

Surprisingly, *for all three datasets* we find that the best-performing method overall is either OCR+PAGE1 or OCR+PAGEN.

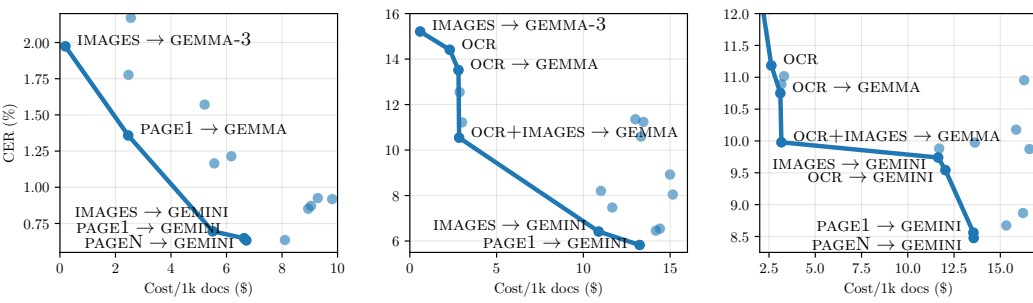

Figure 4: Performance against cost for all methods on IAM (left), Malvern-Hills (center) and Bentham (right). Some outlier points are cut off for clarity. We see that OCR+PAGE1 and OCR+PAGEN are Pareto frontier methods for all three tasks, and the best-performing overall.

Table 1: Various transcription methods on the `IAM` dataset with Azure OCR. 'Minor/Major#Err.' are the average number of trivial and non-trivial errors per document respectively for the given method, as assessed by an LLM; see Section 5.2.1 for details. Our methods are bold; top scores are emphasized: **first**, *second*, third.

| | Input | CER ↓ (%) | Cost ($) /1k docs | Minor#Err. | Major#Err. |
|---|---|---|---|---|---|
| — | OCR | 3.81 | 2.26 | 3.61 | 5.11 |
| GEMMA-3-27B | OCR | 2.93 | 2.42 | 3.97 | 3.17 |
| | IMAGES | 2.31 | 0.19 | 2.65 | 4.32 |
| | **OCR+PAGEN** | 2.17 | 2.55 | 3.44 | 2.22 |
| | IMAGES-ALL-AT-ONCE | 1.97 | 0.19 | 3.45 | 4.05 |
| | OCR+IMAGES | 1.78 | 2.47 | 2.78 | 2.66 |
| | **OCR+PAGE1** | 1.36 | 2.46 | 2.41 | 2.03 |
| GPT-4O | OCR | 1.21 | 6.17 | 2.81 | 1.86 |
| | IMAGES-ALL-AT-ONCE | 0.92 | 9.29 | 1.82 | 1.59 |
| | IMAGES | 0.92 | 9.81 | 2.06 | 1.79 |
| | **OCR+PAGEN** | 0.87 | 9.04 | 1.95 | 1.29 |
| | **OCR+PAGE1** | 0.85 | 8.94 | 1.66 | 1.20 |
| | OCR+IMAGES | 0.72 | 12.69 | 1.86 | 1.21 |
| GEMINI-2.5-PRO | OCR | 1.57 | 5.21 | 1.64 | 1.22 |
| | IMAGES | 1.16 | 5.56 | 1.66 | 1.47 |
| | IMAGES-ALL-AT-ONCE | 0.70 | 5.50 | 1.77 | 1.37 |
| | **OCR+PAGE1** | 0.65 | 6.63 | 2.51 | **1.00** |
| | OCR+IMAGES | *0.64* | 8.10 | 1.63 | 1.32 |
| | **OCR+PAGEN** | **0.63** | 6.71 | 1.80 | *1.04* |

Table 2: Various transcription methods on the `Malvern-Hills` and `Bentham` datasets. The OCR engine is Azure AI Vision. Our methods are bold; top scores are emphasized: **first**, *second*, third.

(a) `Malvern-Hills` dataset

| | Input | CER ↓ (%) | Cost ($) /1k docs |
|---|---|---|---|
| — | OCR | 14.41 | 2.30 |
| GEMMA-3-27B | IMAGES | 27.19 | 0.66 |
| | IMAGES-ALL-AT-ONCE | 15.21 | 0.60 |
| | OCR | 13.52 | 2.80 |
| | **OCR+PAGE1** | 12.55 | 2.87 |
| | **OCR+PAGEN** | 11.22 | 3.01 |
| | OCR+IMAGES | 10.54 | 2.84 |
| GPT-4O | IMAGES-ALL-AT-ONCE | 11.35 | 13.00 |
| | IMAGES | 11.24 | 13.46 |
| | OCR | 10.60 | 13.32 |
| | **OCR+PAGE1** | 8.92 | 15.00 |
| | **OCR+PAGEN** | 8.05 | 15.15 |
| | OCR+IMAGES | 7.11 | 17.54 |
| GEMINI-2.5-PRO | IMAGES-ALL-AT-ONCE | 8.20 | 11.01 |
| | OCR | 7.47 | 11.67 |
| | **OCR+PAGEN** | 6.54 | 14.41 |
| | OCR+IMAGES | 6.46 | 14.18 |
| | IMAGES | *6.42* | 10.88 |
| | **OCR+PAGE1** | **5.83** | 13.24 |

(b) Bentham dataset

| | Input | CER ↓ (%) | Cost ($) /1k docs |
|---|---|---|---|
| — | OCR | 11.18 | 2.63 |
| GEMMA-3-27B | IMAGES-ALL-AT-ONCE | 27.88 | 0.50 |
| | IMAGES | 15.12 | 0.59 |
| | **OCR+PAGEN** | 11.02 | 3.31 |
| | **OCR+PAGE1** | 10.89 | 3.17 |
| | OCR | 10.75 | 3.11 |
| | OCR+IMAGES | 9.98 | 3.17 |
| GPT-4O | OCR+PAGE1 | 10.95 | 16.29 |
| | IMAGES-ALL-AT-ONCE | 10.18 | 15.85 |
| | OCR | 9.97 | 13.63 |
| | IMAGES | 9.87 | 16.58 |
| | OCR+IMAGES | 9.35 | 20.93 |
| | **OCR+PAGEN** | 8.87 | 16.23 |
| GEMINI-2.5-PRO | IMAGES-ALL-AT-ONCE | 9.88 | 11.70 |
| | IMAGES | 9.74 | 11.64 |
| | OCR | 9.54 | 12.03 |
| | OCR+IMAGES | 8.67 | 15.32 |
| | **OCR+PAGE1** | *8.56* | 13.55 |
| | **OCR+PAGEN** | **8.48** | 13.56 |

Both of these methods have access to *less than half*, and sometimes as little as a quarter, of the original source data, the document images. Yet they outperform methods that have access to all of the images, and those with access to both the OCR text *and* all of the images, despite the fact that *the performance of OCR **alone** is poor for all three tasks* — for IAM it is the worst method outright, and for Malvern-Hills and Bentham it is outperformed by all GEMINI- and GPT-4O-based models.

The performance gap in each case is not large, but our intention is not to propose a SOTA method. It is to demonstrate that MLLMs can leverage common context from limited, expensive image input to improve correction of cheap text input. Put simply, a single image is often as good as the full document. Existing simplistic approaches to transcription based solely on OCR (with traditional OCR engines or MLLMs end-to-end), miss useful context, and methods that use both OCR and all images can overwhelm a model with redundant repeated context that can be found in a single image.

**Semantic accuracy.** Table 1 includes additional columns with information about document error types; this is described in Section 5.2.1. Counting MLLM errors semantically, rather than with strict CER, we see that the performance gap between OCR+PAGE1 and OCR+PAGEN and the next best method, OCR+IMAGES→GEMINI-2.5-PRO, is even larger. While all three methods have a similar average number of minor errors per document (semantic and formatting errors that do not affect meaning), our methods have over 20% fewer major errors (genuine mistakes or hallucinations) than the next best method.

## 6 LIMITATIONS AND FUTURE WORK

We stated that we would expect OCR+PAGEN to be at least as good as OCR+PAGE1, yet in practice we find that neither method is consistently better than the other across datasets or MLLMs. We discuss possible reasons and solutions for this in Appendix C.2.

**Cost and scaling.** Although our methods use only a single page, they are sometimes more expensive than methods which use the full set of images. This is due to a combination of high text-density OCR transcriptions, as well as the added cost of OCR transcription itself. One way to address this could be the use of cheaper OCR; we note that some MLLMs, such as GEMMA-3-27B, are significantly cheaper than our chosen OCR engine, and can achieve improved or comparable results over OCR depending on the task. Further experimentation using cheap MLLMs as alternative OCR engines and more powerful MLLMs as post-processors is an area we leave for future work.

We also note that for our tasks the average number of pages per document is between 2 and 3. Our methods use a single image regardless of document length, so we might expect more efficient cost scaling as document length increases. On the other hand, performance may suffer from the added challenge of processing longer documents with longer contexts in a single pass. One potential way to leverage the scaling benefits of OCR+PAGE1 while keeping context size small could be prompt caching (Shi et al., 2024). Cached tokens are typically much cheaper for commercial MLLMs. Therefore, a 'page-by-page' version of OCR+PAGE1 could use the same page image to provide context to smaller chunks (e.g. a few pages at a time) of a long document, and the cost would remain low as the image tokens for the context image would be cached — similar to in-context examples, which also benefit from prompt caching. We leave this investigation for future work.

**Conclusion.** In this work we investigated the transcription of multi-page handwritten documents using various configurations of commercial OCR engines and MLLMs. We provide a set of multi-page transcription benchmarks, including a brand new dataset, Malvern-Hills, which we hope will serve as a useful evaluation tool for the community. We also provide the first known evaluation of the effectiveness of different prompting strategies for the task of zero-shot, multi-page handwriting transcription, on our three benchmark tasks. We propose the OCR+PAGE1 and OCR+PAGEN methods, and empirically demonstrate that they improve transcription accuracy while balancing cost and performance. Notably, they are equally or more effective than end-to-end processing with leading MLLMs, despite not having access to all page images.

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

## A    EXAMPLE DOCUMENTS FROM DATASETS

Figures 5, 6 & 7 show example pages from the `IAM`, `Malvern-Hills` and `Bentham` datasets respectively.

(a) An example of a document from the IAM Handwriting Database.

(b) An example of a constructed multi-page `IAM` document, combining two pages from the same writer with the machine-printed text cropped out.

Figure 5: Left: original IAM document. Right: constructed two-page `IAM` document.

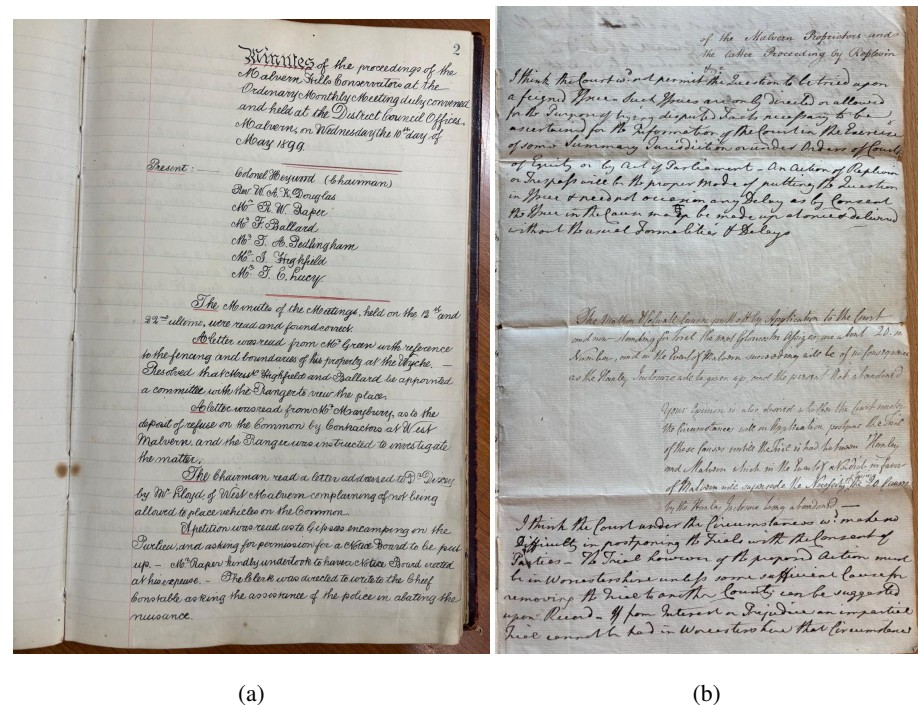

Figure 6: Example pages from the `Malvern-Hills` dataset.

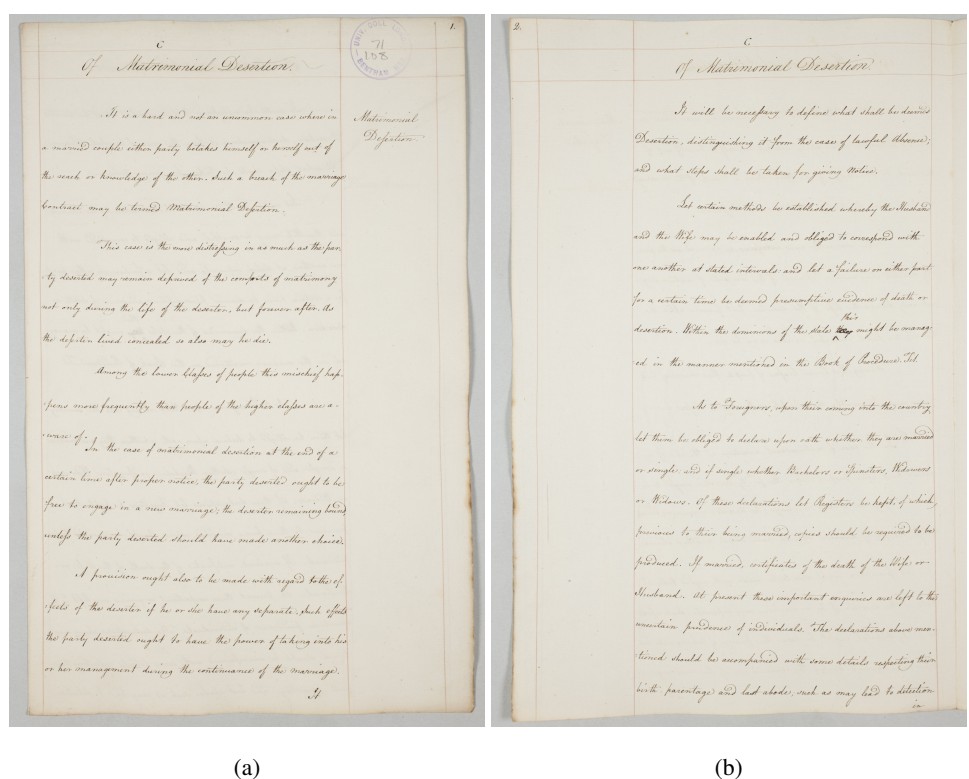

Figure 7: Example pages from the `Bentham` dataset.

# B    EXAMPLES OF OCR+PAGE1 CORRECTIONS

See Figures 8–11. All examples are on IAM, use Google Cloud Vision as the OCR engine and GPT-4O.

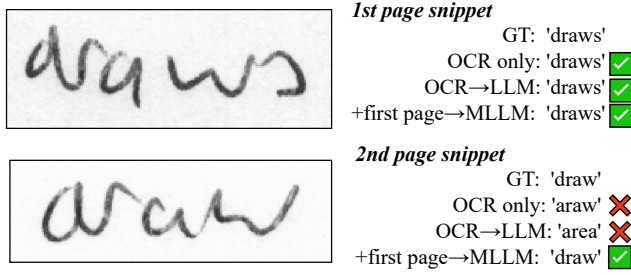

Figure 8: With OCR+PAGE1, the correctly-transcribed occurrence of 'draws' in the first page can be extrapolated to the unseen 'draw' on the second page.

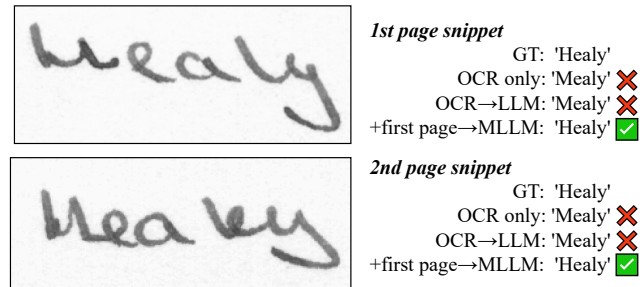

Figure 9: With OCR+PAGE1, the correctly-transcribed occurrence of the name 'Mr Healy' in the first page can be extrapolated to the unseen occurence on the second page.

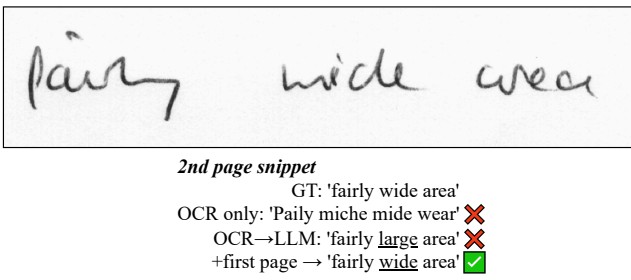

Figure 10: OCR+PAGE1 corrects where OCR→LLM gets it wrong, despite *only having access only to the garbled OCR output* and not the image of the word 'wide' shown above. Suggests some degree of reasoning using the seemingly irrelevant first page text — i.e. it can see that 'm's on page 1 look similar to 'w's and reason that 'mide' could be 'wide'.

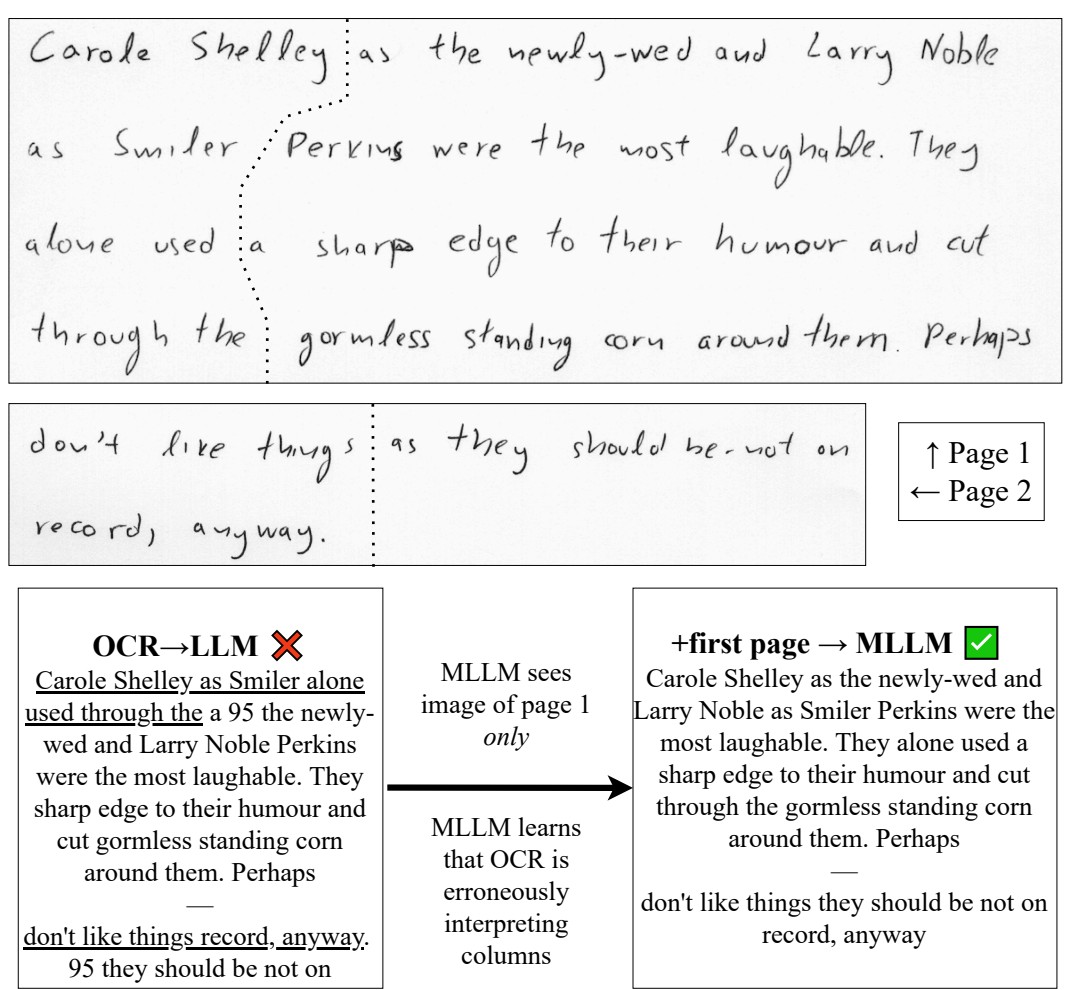

Figure 11: An unusual case where the OCR engine erroneously breaks the text into two columns for both pages. This is shown using the dotted line and underlining of the transcribed text. OCR→LLM preserves this error. OCR+PAGE1 trivially corrects it in the first page — it has access to the image — *but also corrects it in the second page*, even though it has no access to that image. OCR+PAGE1 correctly rearranges the text on page 2 using only context and the inferred formatting from page 1.

# C  ADDITIONAL RESULTS AND DISCUSSION

Tables 3, 4 & 5 show results of early experiments with GPT-4O and GPT-4O-MINI for earlier prompt iterations of our methods, on a multi-page validation split derived from the IAM Database. Each table uses a different baseline OCR engine: Azure, Google Cloud Vision and Amazon Textract. As Azure performs the best of the three, and is the cheapest (see Table 7), we use it for final experiments in the main text. We also tested Tesseract, but found it to be completely incapable of producing any meaningful transcription of handwritten text.

Table 3: IAM: relative performance of transcription methods. Rel(ative) imp(rovement) is against the baseline OCR (Azure), and cost is for processing the entire dataset with the given method.

| Method | → MLLM | CER | Rel. Imp. | Cost ($) |
|---|---|---|---|---|
| OCR | - | 0.036 | 0.00 | 0.59 |
| OCR | GPT-4O-MINI | 0.032 | 0.11 | 0.64 |
| OCR | GPT-4O | 0.033 | 0.08 | 1.42 |
| OCR-PBP | GPT-4O-MINI | 0.025 | 0.31 | 0.65 |
| OCR-PBP | GPT-4O | 0.029 | 0.21 | 1.50 |
| OCR+PAGEN | GPT-4O-MINI | 0.029 | 0.19 | 2.12 |
| OCR+PAGEN | GPT-4O | 0.025 | 0.29 | 2.24 |
| VISION | GPT-4O | 0.027 | 0.24 | 2.32 |
| VISION-PBP | GPT-4O | **0.010** | 0.73 | 2.43 |
| OCR+ALL-PAGES | GPT-4O | 0.027 | 0.24 | 3.10 |
| OCR+ALL-PAGES-PBP | GPT-4O | 0.011 | 0.68 | 3.24 |
| ALL-OCR-PBP | GPT-4O-MINI | 0.020 | 0.46 | 2.48 |
| ALL-OCR-PBP | GPT-4O | 0.021 | 0.43 | 4.07 |
| OCR+PAGE1 | GPT-4O-MINI | *0.015* | 0.59 | 2.10 |
| OCR+PAGE1 | GPT-4O | 0.027 | 0.26 | 2.20 |

Table 4: Relative performance of MLLMs and prompting strategies compared to the baseline **Google** OCR engine on the IAM dataset.

| Method | → MLLM | CER | Rel. Imp. | Cost ($) |
|---|---|---|---|---|
| OCR | - | 0.095 | 0.00 | 0.89 |
| OCR | GPT-4O-MINI | 0.074 | 0.23 | 0.94 |
| OCR-PBP | GPT-4O-MINI | 0.071 | 0.26 | 0.95 |
| OCR | GPT-4O | 0.064 | 0.33 | 1.73 |
| OCR-PBP | GPT-4O | 0.064 | 0.33 | 1.81 |
| VISION | GPT-4O | 0.027 | 0.71 | 2.32 |
| OCR+PAGE1 | GPT-4O-MINI | 0.047 | 0.51 | 2.40 |
| OCR+PAGEN | GPT-4O-MINI | 0.060 | 0.37 | 2.40 |
| VISION-PBP | GPT-4O | **0.010** | 0.90 | 2.43 |
| ALL-OCR-PBP | GPT-4O-MINI | 0.020 | 0.80 | 2.48 |
| OCR+PAGE1 | GPT-4O | 0.042 | 0.56 | 2.50 |
| OCR+PAGEN | GPT-4O | 0.044 | 0.54 | 2.53 |
| OCR+ALL-PAGES | GPT-4O | 0.035 | 0.63 | 3.39 |
| OCR+ALL-PAGES-PBP | GPT-4O | *0.019* | 0.80 | 3.54 |
| ALL-OCR-PBP | GPT-4O | 0.021 | 0.78 | 4.07 |

Table 5: Relative performance of MLLMs and prompting strategies compared to the baseline **Textract** OCR engine on the `IAM` dataset.

| Method | → MLLM | CER | Rel. Imp. | Cost ($) |
|---:|---:|---:|---:|---:|
| OCR | - | 0.050 | 0.00 | 0.89 |
| OCR | GPT-4O-MINI | 0.051 | -0.03 | 0.94 |
| OCR-PBP | GPT-4O-MINI | 0.045 | 0.10 | 0.95 |
| OCR | GPT-4O | 0.046 | 0.08 | 1.71 |
| OCR-PBP | GPT-4O | 0.041 | 0.18 | 1.79 |
| VISION | GPT-4O | 0.027 | 0.45 | 2.32 |
| OCR+PAGE1 | GPT-4O-MINI | 0.027 | 0.45 | 2.40 |
| OCR+PAGEN | GPT-4O-MINI | 0.046 | 0.08 | 2.41 |
| VISION-PBP | GPT-4O | **0.010** | 0.81 | 2.43 |
| ALL-OCR-PBP | GPT-4O-MINI | 0.020 | 0.61 | 2.48 |
| OCR+PAGE1 | GPT-4O | 0.030 | 0.40 | 2.50 |
| OCR+PAGEN | GPT-4O | 0.027 | 0.46 | 2.54 |
| OCR+ALL-PAGES | GPT-4O | 0.029 | 0.42 | 3.39 |
| OCR+ALL-PAGES-PBP | GPT-4O | *0.011* | 0.78 | 3.54 |
| ALL-OCR-PBP | GPT-4O | 0.021 | 0.59 | 4.07 |

## C.1 SEMANTIC EVALUATION OF TRANSCRIPTION ACCURACY WITH AN LLM

Table 6 shows the full set of error classifications for `IAM`, as described in Section 5.2.1 and summarised in Table 1. For conciseness, the table combines the 'proper noun' and 'numerical' error types into one column, and combines the two minor error types, 'formatting' and 'semantic', into the 'minor' error type. The error types are defined in the prompt used to extract them as follows:

```
Error types (choose exactly one per error):
1) missing_content
   - A deletion with no suitable inserted counterpart.
   Example: 'word' with no matching 'word'.
   - Minor missing content, such as single punctuation marks,
   should be 'formatting'.
2) hallucination
   - An insertion with no corresponding gt (including numbers
   that appear only in pred).
   - Minor hallucinations, such as single punctuation marks,
   should be 'formatting'.
3) mistake_proper_noun
   - gt is a proper noun (possibly multi-word: full or partial
   names, places, institutions, titles) and pred misrenders it
   (letters/wording/order).
   - Pure capitalization changes that don't create a different name
   are semantic.
   - Extended 'proper names', such as full names, should be
   treated in their entirety, e.g. 'A. B. Smith'
4) mistake_numerical
   - Use only when gt contains a number/date/roman numeral/measure
   and pred changes its value or structure or fails to include it.
   - Combine obviously linked components (e.g., an entire date
   like 18 March 1884) into one numerical error.
5) mistake_other
   - Non{proper-noun, non-numeric word/phrase error that affects
   normal reading (e.g., genuine misspelling or wrong lemma/word).
   - Use sparingly, only if none of the other error types are
   appropriate, not as a catch-all
```

```
6) semantic
  - Differences that are clearly meaning-preserving or trivial:
    - minor letter/spelling differences where the word couldn't
    reasonably be mistaken for another,
    - alternative/modernized spellings or contractions with
    the same meaning
    - capitalization-only changes,
7) formatting
  - Minor erroneous punctuation, misplaced or missing newlines,
  misplaced structural content including text that has been moved
  from one place in the transcription to another
  - If a moved block contains internal real mistakes (e.g., a
  wrong year), add separate errors for those internal pieces with
  the appropriate types.
```

## C.2 WHY ISN'T OCR+PAGEN ALWAYS BETTER THAN OCR+PAGE1?

While OCR+PAGEN often performs better than OCR+PAGE1, this is not consistent over MLLMs or datasets. We attribute this to the added prompt complexity of extrapolating from an arbitrary Nth page rather than the 1st, as this is the only difference between the two methods. There is evidence that MLLMs are sensitive to prompt order (Guan et al., 2025), so it is not unreasonable to suppose that the prompt ordering for OCR+PAGE1, which is 'page 1 image, page 1 OCR text, page 2 OCR text, ...', where the corresponding image and text are adjacent in the prompt, is easier for an MLLM to follow than 'page N image, page 1 OCR text, ..., page N OCR text, ...'. The method relies on learning a mapping from the page N image to the page N OCR, and MLLMs are known to be better at local context reasoning than long-context (Liu et al., 2025). In the case where the first page is approximately as informative as the Nth, the complexity this prompt arrangement introduces may outweigh any marginal benefit from the page ID choice. It is possible that further prompt tuning, such as rearranging the image within the prompt, could mitigate this; we leave it for future work.

## D ADDITIONAL RESOURCES

Below are links to several online tools mentioned in this paper:

- Tesseract: `https://github.com/tesseract-ocr/tesseract`
- LLM-Aided OCR: `https://github.com/Dicklesworthstone/llm_aided_ocr`
- BetterOCR: `https://github.com/junhoyeo/BetterOCR`

**Figures.** We acknowledge the use of (both original and edited) open-licensed SVG vectors from SVG Repo:

- Images clipart by Ionicons: `https://www.svgrepo.com/svg/327088/images-sharp`
- Robot clipart by Konstantin Filatov `https://www.svgrepo.com/svg/521818/robot`
- Documents clipart by SVG repo: `https://www.svgrepo.com/svg/139884/documents-papers`

**Malvern-Hills dataset.** We are grateful to the Malvern Hills Trust for providing images.

## D.1 COMMERCIAL TOOLS

Costs of commercial OCR engines and LLMs for estimates used in this paper are given in Tables 7, 8. Most have pricing tiers based on scale, or a limited free allowance; for simplicity we take the lowest (non-batch) cost per run for each engine, and for the OpenAI API.

Table 6: IAM error types for all runs in Table 1, evaluated by GEMINI-2.5-FLASH. Each cell contains the average number of occurrences of that type of error in a transcription produced by the given method, and the percentage of total errors in a transcription that are that particular error type. E.g., Azure OCR output alone has an average of 5.1 major errors per transcription for IAM documents, but 41.4% of total errors inn Azure transcription are minor, i.e. semantic or formatting-related only.

| | Input | Major (all) | Missing content | Halluc -ination | Proper noun /Numeric | Other error | Minor error |
|---|---|---|---|---|---|---|---|
| — | OCR | 5.1 (58.6%) | 0.3 (3.1%) | 0.2 (1.9%) | 0.8 (9.0%) | 3.9 (44.6%) | 3.6 (41.4%) |
| GEMMA-3-27B | IMAGES | 4.3 (52.1%) | 0.2 (2.7%) | 0.2 (1.9%) | 1.1 (13.0%) | 2.9 (34.5%) | 4.0 (47.9%) |
| | IMAGES-ALL-AT-ONCE | 4.0 (60.4%) | 0.2 (3.3%) | 0.1 (1.8%) | 0.9 (13.8%) | 2.8 (41.4%) | 2.7 (39.6%) |
| | OCR | 3.2 (47.9%) | 0.3 (4.2%) | 0.2 (2.4%) | 0.6 (8.8%) | 2.1 (32.5%) | 3.4 (52.1%) |
| | OCR+IMAGES | 2.7 (43.6%) | 0.2 (3.5%) | 0.1 (1.5%) | 0.5 (8.2%) | 1.9 (30.3%) | 3.4 (56.4%) |
| | OCR+PAGEN | 2.2 (44.5%) | 0.2 (4.7%) | 0.1 (3.0%) | 0.5 (9.7%) | 1.4 (27.1%) | 2.8 (55.5%) |
| | OCR+PAGE1 | 2.0 (45.7%) | 0.3 (5.7%) | 0.1 (3.2%) | 0.4 (8.4%) | 1.3 (28.4%) | 2.4 (54.3%) |
| GPT-4O | OCR | 1.9 (39.8%) | 0.2 (4.4%) | **0.0** **(1.0%)** | 0.5 (10.4%) | 1.1 (24.0%) | 2.8 (60.2%) |
| | IMAGES | 1.8 (49.6%) | 0.2 (5.7%) | 0.1 (1.6%) | 0.5 (13.2%) | 1.1 (29.2%) | 1.8 (50.4%) |
| | IMAGES-ALL-AT-ONCE | 1.6 (43.6%) | 0.2 (4.9%) | 0.1 (1.5%) | 0.4 (11.3%) | 0.9 (25.9%) | 2.1 (56.4%) |
| | OCR+PAGEN | 1.3 (39.8%) | 0.2 (5.5%) | **0.0** **(0.9%)** | 0.3 (7.8%) | 0.8 (25.6%) | 2.0 (60.2%) |
| | OCR+IMAGES | *1.2* *(42.2%)* | 0.2 (7.1%) | **0.0** **(1.3%)** | 0.3 (8.7%) | 0.7 (25.0%) | 1.7 (57.8%) |
| | OCR+PAGE1 | *1.2* *(39.1%)* | **0.1** **(4.9%)** | **0.0** **(1.5%)** | 0.3 (8.6%) | 0.7 (24.2%) | 1.9 (60.9%) |
| GEMINI-2.5-PRO | IMAGES | 1.5 (47.1%) | 0.2 (6.6%) | 0.2 (5.4%) | 0.3 (8.1%) | 0.8 (27.0%) | **1.6** **(52.9%)** |
| | IMAGES-ALL-AT-ONCE | 1.4 (45.2%) | 0.2 (7.4%) | 0.1 (2.8%) | 0.2 (8.0%) | 0.8 (27.1%) | 1.7 (54.8%) |
| | OCR+IMAGES | 1.3 (42.7%) | 0.2 (6.7%) | 0.1 (3.9%) | 0.2 (6.7%) | 0.8 (25.5%) | 1.8 (57.3%) |
| | OCR | *1.2* *(32.8%)* | **0.1** **(4.0%)** | 0.1 (2.8%) | 0.2 (6.3%) | 0.7 (19.8%) | 2.5 (67.2%) |
| | OCR+PAGEN | **1.0** **(38.9%)** | 0.2 (7.7%) | 0.1 (2.5%) | **0.1** **(4.6%)** | **0.6** **(24.2%)** | **1.6** **(61.1%)** |
| | OCR+PAGE1 | **1.0** **(35.7%)** | 0.2 (6.7%) | 0.1 (2.7%) | **0.1** **(5.0%)** | **0.6** **(21.3%)** | 1.8 (64.3%) |

Costs for each tool were taken from their respective webpages:

- Microsoft Azure: https://azure.microsoft.com/en-gb/pricing/details/cognitive-services/computer-vision/
- Amazon Textract: https://aws.amazon.com/textract/pricing/
- Google Cloud Vision: https://cloud.google.com/vision/pricing
- OpenAI: https://openai.com/api/pricing/

- Google Gemini: `https://ai.google.dev/gemini-api/docs/pricing`
- Google does not provide pricing information for GEMMA-3-27B, and only provide it for free with usage limits. For this reason we take our estimate of the price of GEMMA-3-27B from `https://openrouter.ai/google/gemma-3-27b-it`

Table 7: Pricing for OCR Engines

| OCR Engine | Cost per 1k Calls ($) |
|---|---|
| Azure AI Vision | 1.00 |
| Google Cloud Vision | 1.50 |
| Amazon Textract | 1.50 |

Table 8: Pricing for LLMs

| LLM | Cost per 1M Tokens ($) | |
|---|---|---|
| | Input | Output |
| GEMMA-3-27B | 0.07 | 0.50 |
| GPT-4O | 2.50 | 10.00 |
| GEMINI-2.5-PRO | 1.25 | 10.00 |

## E    REBUTTAL EXPERIMENTS

This section contains new experiments and data obtained during the ICLR review period. In the final version these details will be incorporated into the main paper or existing/new Appendix sections as appropriate. The code repo will be updated to include code required to reproduce all additional experiments.

**Results on document datasets with longer page counts (5-13 pages).** We re-work `IAM` and `Malvern-Hills` to generate three longer-document benchmarks: `IAM-5`, with 5 pages per document, `Malvern-Hills-5+`, with an average of 5.8 pages per document, and `Malvern-Hills-10+`, with an average of 11.5 pages per document. We also introduce a new Chinese handwritten benchmark with 5 page documents; see below.

**Results on an additional, non-Latin dataset.** We introduce `casia-5`, a dataset of generated 5 page documents derived from CASIA HDB2.1Test; a Chinese-language handwritten dataset (Liu et al., 2011)

**Results on three additional methods that are not commercial MLLMs.** These are:

- PYLAIA (Tarride et al., 2024): a CNN-RNN-CTC recognizer trained on IAM-DB with lexicon-aware decoding and an n-gram language model. We evaluate PYLAIA on `IAM`.
- TROCR (Li et al., 2023): specifically `trocr-large-handwritten`[3], a pre-trained Transformer model fine-tuned on handwritten text. We evaluate TROCR on `Malvern-Hills-5+`.
- DOCOWL2[4] (Hu et al., 2025): an open MLLM specialized for document processing, which we evaluate on `Malvern-Hills-5+`

**Results on a new method, OCR+PAGER** OCR+PAGER is a middle-ground method between OCR+PAGE1 and OCR+PAGEN which provides a *random* page image to the post-processing MLLM, rather than exclusively the first page, or a page chosen by another LLM. The post-processing prompt is the same as for OCR+PAGEN.

**An ablation on a randomized 5-page version of IAM: `IAM-5-Random`,** to demonstrate the case when multi-page documents share neither semantic content or a common author.

**Detailed data from experiments, including** token breakdowns per method by image, input text and output text, cost breakdowns per method by OCR, MLLM input and MLLM output, inference times, and MLLM failure rates. All API calls are made in series (not batch); see code repo for implementation.

We also include additional metadata for the `Malvern-Hills` dataset in Tables 15–18.

### E.1    CASIA-5, A CHINESE HANDWRITTEN DATASET

We take the CASIA-HWDB2.1 test split, which consists of 300 images of multiple lines of handwritten Chinese text. Similar to the process for IAM, we generate multi-page dataset `casia-5` by randomly grouping pages with the same author ID, such that handwriting, but not semantic content, is consistent. We use a subset of 30 documents of 5 pages each in our experiments. We use Google Cloud Vision as our OCR engine.

See results in Table 9.

**Discussion.** For this challenging Chinese handwriting dataset, we see that the best-performing model overall is OCR+IMAGES, i.e. a model that has access to both OCR prediction and the full set of document images. The OCR engine on its own is already quite effective, indeed, no method

---

[3]`https://huggingface.co/microsoft/trocr-large-handwritten`
[4]`https://huggingface.co/mPLUG/DocOwl2`

Table 9: Results on `casia-5` for all methods described in the main text, plus OCR+PAGER. Our methods, i.e. OCR plus single-page methods, are bolded, and the best-/second-best-performing scores are bolded/italicized. 'Fails (/doc)' is the average number of retried API calls required per method (all calls eventually succeeded). For GPT-4O, all retries were due to incomplete/truncated response errors. For GEMINI-2.5-PRO, retries were due to invalid JSON output errors.

| | Method | CER ↓ (%) | Cost ($/1k docs) Total | OCR | In | Out | Tokens (/doc) Image | In | Out | Time (s/doc) | Fails (/doc) |
|---|---|---|---|---|---|---|---|---|---|---|---|
| — | OCR | 12.84 | 7.50 | 7.50 | 0.00 | 0.00 | 0 | 0 | 0 | — | — |
| GPT-4O | IMAGES-AAO | 43.64 | 29.30 | 0.00 | 14.86 | 14.44 | 5,525 | 252 | 1,444 | 74.8 | 0.10 |
| | IMAGES | 42.40 | 30.81 | 0.00 | 16.91 | 13.90 | 5,525 | 1,180 | 1,389 | 84.4 | 0.03 |
| | OCR | 18.32 | 26.75 | 7.50 | 6.13 | 13.12 | 55 | 2,395 | 1,311 | 36.6 | 0.00 |
| | **OCR+PAGE1** | 14.48 | 28.31 | 7.50 | 7.45 | 13.36 | 1,105 | 1,741 | 1,335 | 44.2 | 0.00 |
| | OCR+IMAGES | 14.16 | 41.58 | 7.50 | 20.58 | 13.50 | 5,580 | 2,650 | 1,350 | 78.5 | 0.00 |
| | **OCR+PAGER** | 13.91 | 28.41 | 7.50 | 7.51 | 13.40 | 1,105 | 1,765 | 1,340 | 34.0 | 0.00 |
| | **OCR+PAGEN** | 12.87 | 28.68 | 7.50 | 7.64 | 13.55 | 1,105 | 1,765 | 1,350 | 47.7 | 0.00 |
| GEMINI | OCR | 33.17 | 24.87 | 7.50 | 2.81 | 14.56 | 0 | 2,249 | 1,455 | 30.7 | 0.00 |
| | IMAGES-AAO | 23.07 | 18.09 | 0.00 | 5.32 | 12.77 | 3,995 | 262 | 1,277 | 18.8 | 0.00 |
| | IMAGES | 13.55 | 19.08 | 0.00 | 6.51 | 12.58 | 3,995 | 1,210 | 1,257 | 31.1 | 0.00 |
| | **OCR+PAGE1** | 12.61 | 23.25 | 7.50 | 3.05 | 12.70 | 799 | 1,637 | 1,270 | 48.3 | 0.07 |
| | **OCR+PAGER** | 10.13 | 23.19 | 7.50 | 3.08 | 12.61 | 799 | 1,662 | 1,261 | 18.0 | 0.00 |
| | **OCR+PAGEN** | *8.94* | 23.42 | 7.50 | 3.20 | 12.72 | 799 | 1,662 | 1,267 | 27.5 | 0.00 |
| | OCR+IMAGES | **8.39** | 27.80 | 7.50 | 8.14 | 12.16 | 3,995 | 2,519 | 1,215 | 40.2 | 0.00 |

involving GPT-4o improves on it, and many significantly worsen it. The only methods that improve on OCR alone are those that use both OCR and at least one image as input to GEMINI. The fact that OCR+PAGEN and OCR+IMAGES provide a similar performance boost over OCR alone supports our findings that a single well-chosen image can provide most of the benefit provided by the full set of images, and at a lower token/overall cost.

## E.2 `IAM-5`

We generate 5-page documents by the same method we generated `IAM`; by grouping random individual pages by author ID so handwriting (but not necessarily semantic content) is consistent. We use 100 page images to generate 20 5-page documents.

See results in Table 10.

Table 10: Results on `IAM-5` for all methods described in the main text, plus OCR+PAGER. The single method with failed calls was due to GEMMA 'free resource exhausted' errors.

| | Method | CER ↓ | Cost ($/1k docs) | | | | Tokens (/doc) | | | Time | Fails |
| | | | Total | OCR | In | Out | Image | In | Out | (s/doc) | (/doc) |
|---|---|---|---|---|---|---|---|---|---|---|---|
| — | OCR | 3.36 | 5.00 | 5.00 | 0.00 | 0.00 | 0 | 0 | 0 | — | — |
| GEMMA | **OCR+PAGER** | 2.69 | 5.40 | 5.00 | 0.13 | 0.27 | 886 | 939 | 537 | 12.2 | 0.00 |
| | OCR | 2.67 | 5.34 | 5.00 | 0.11 | 0.23 | 0 | 1,543 | 467 | 14.8 | 0.00 |
| | **OCR+PAGEN** | 2.52 | 5.51 | 5.00 | 0.20 | 0.31 | 874 | 939 | 535 | 18.3 | 0.00 |
| | IMAGES | 1.54 | 0.65 | 0.00 | 0.40 | 0.26 | 4,436 | 1,210 | 513 | 20.1 | 0.00 |
| | OCR+IMAGES | 1.47 | 5.68 | 5.00 | 0.44 | 0.24 | 4,436 | 1,813 | 480 | 19.5 | 0.05 |
| | IMAGES-AAO | 1.35 | 0.59 | 0.00 | 0.33 | 0.26 | 4,436 | 262 | 525 | 12.7 | 0.00 |
| | **OCR+PAGE1** | 1.22 | 5.39 | 5.00 | 0.12 | 0.27 | 856 | 914 | 531 | 12.3 | 0.00 |
| GPT-4O | OCR | 0.99 | 13.60 | 5.00 | 3.91 | 4.69 | 58 | 1,503 | 469 | 21.6 | 0.00 |
| | **OCR+PAGE1** | 0.98 | 14.82 | 5.00 | 5.01 | 4.81 | 1,003 | 867 | 481 | 24.3 | 0.00 |
| | **OCR+PAGER** | 0.91 | 14.93 | 5.00 | 5.11 | 4.82 | 1,020 | 891 | 482 | 21.9 | 0.00 |
| | **OCR+PAGEN** | 0.83 | 15.05 | 5.00 | 5.19 | 4.87 | 1,020 | 891 | 482 | 26.6 | 0.00 |
| | IMAGES-AAO | 0.77 | 18.60 | 0.00 | 13.88 | 4.72 | 5,134 | 252 | 472 | 38.2 | 0.00 |
| | OCR+IMAGES | 0.58 | 27.04 | 5.00 | 17.38 | 4.66 | 5,192 | 1,758 | 466 | 62.2 | 0.00 |
| | IMAGES | 0.54 | 20.68 | 0.00 | 15.94 | 4.75 | 5,134 | 1,180 | 474 | 60.6 | 0.00 |
| GEMINI | OCR | 1.37 | 11.51 | 5.00 | 1.93 | 4.58 | 0 | 1,543 | 457 | 11.3 | 0.00 |
| | IMAGES-AAO | 0.54 | 11.18 | 0.00 | 5.87 | 5.31 | 4,436 | 262 | 530 | 7.2 | 0.00 |
| | IMAGES | 0.52 | 11.72 | 0.00 | 7.06 | 4.67 | 4,436 | 1,210 | 466 | 21.4 | 0.00 |
| | **OCR+PAGER** | 0.52 | 12.48 | 5.00 | 2.28 | 5.20 | 886 | 939 | 520 | 8.3 | 0.00 |
| | **OCR+PAGEN** | *0.51* | 12.59 | 5.00 | 2.34 | 5.25 | 874 | 939 | 520 | 13.5 | 0.00 |
| | OCR+IMAGES | **0.47** | 17.39 | 5.00 | 7.81 | 4.57 | 4,436 | 1,813 | 457 | 18.1 | 0.00 |
| | **OCR+PAGE1** | **0.47** | 12.40 | 5.00 | 2.21 | 5.19 | 856 | 914 | 518 | 7.9 | 0.00 |

**Discussion.** We see similar results for the longer document case of `IAM-5` as we did for `IAM`. OCR+PAGE1 remains the (joint) top-performing model and all OCR + single-image methods perform approximately as well as the full OCR+IMAGES method, despite lacking access to *80% of images* (and the comparatively poor performance of OCR alone), and correspondingly have lower costs and inference times.

## E.3 `IAM-5-RANDOM`; INCONSISTENT HANDWRITING *and* SEMANTIC CONTENT ABLATION

`IAM-5-Random` is generated in the same way as `IAM-5`, but we ensure that all 20 generated documents contain pages from 5 different authors.

See results in Table 11.

**Discussion.** As expected, the benefit of OCR+single-image methods is less pronounced when there is neither consistent handwriting nor semantic content across pages — *but there is still some benefit*. Though OCR+PAGE1 underperforms, +PAGER and +PAGEN are perform comparably with methods that include access to all images, at lower cost and faster inference time. These results demonstrate, firstly, that our intuition about the benefit of +PAGEX methods is likely correct; the more similar pages in a document are (in content, writing, etc.) the more performance gain can be achieved with only a single page (and vice versa); secondly, that even in cases where pages are quite different, a single page can still provide performance benefit. This may be a result of other page

Table 11: Results on `IAM-5-Random` for all methods described in the main text, plus OCR+PAGER.

| | Method | CER ↓ | Cost ($/1k docs) | | | | Tokens (/doc) | | | Time (s/doc) | Fails (/doc) |
|---|---|---|---|---|---|---|---|---|---|---|---|
| | | | Total | OCR | In | Out | Image | In | Out | | |
| — | OCR | 3.78 | 5.00 | 5.00 | — | — | — | — | — | — | — |
| GEMMA | **OCR+PAGER** | 3.14 | 5.40 | 5.00 | 0.13 | 0.27 | 914 | 942 | 537 | 11.8 | 0.0 |
| | **OCR+PAGEN** | 3.10 | 5.51 | 5.00 | 0.20 | 0.31 | 902 | 942 | 540 | 18.0 | 0.0 |
| | OCR | 2.88 | 5.34 | 5.00 | 0.11 | 0.23 | 0 | 1,547 | 468 | 13.8 | 0.0 |
| | IMAGES | 2.34 | 0.67 | 0.00 | 0.41 | 0.26 | 4,671 | 1,210 | 514 | 20.7 | 0.0 |
| | IMAGES-AAO | 2.04 | 0.61 | 0.00 | 0.35 | 0.26 | 4,671 | 262 | 526 | 12.6 | 0.0 |
| | OCR+IMAGES | 1.75 | 5.70 | 5.00 | 0.45 | 0.24 | 4,671 | 1,817 | 484 | 17.2 | 0.0 |
| | **OCR+PAGE1** | 1.53 | 5.39 | 5.00 | 0.13 | 0.27 | 909 | 917 | 532 | 11.7 | 0.0 |
| GPT-4O | IMAGES-AAO | 1.20 | 19.36 | 0.00 | 14.60 | 4.76 | 5,457 | 252 | 475 | 35.6 | 0.0 |
| | OCR | 1.09 | 13.66 | 5.00 | 3.92 | 4.73 | 58 | 1,510 | 473 | 21.4 | 0.0 |
| | **OCR+PAGE1** | 1.08 | 15.03 | 5.00 | 5.19 | 4.83 | 1,071 | 872 | 483 | 19.8 | 0.0 |
| | **OCR+PAGER** | 1.01 | 15.05 | 5.00 | 5.21 | 4.84 | 1,054 | 896 | 483 | 20.3 | 0.0 |
| | **OCR+PAGEN** | 0.92 | 15.18 | 5.00 | 5.29 | 4.90 | 1,054 | 896 | 486 | 25.7 | 0.0 |
| | IMAGES | 0.91 | 21.47 | 0.00 | 16.66 | 4.82 | 5,457 | 1,180 | 481 | 56.8 | 0.0 |
| | OCR+IMAGES | *0.71* | 27.83 | 5.00 | 18.12 | 4.71 | 5,481 | 1,765 | 471 | 55.0 | 0.0 |
| GEMINI | OCR | 1.56 | 11.54 | 5.00 | 1.93 | 4.60 | 0 | 1,547 | 460 | 10.9 | 0.0 |
| | **OCR+PAGE1** | 1.31 | 12.49 | 5.00 | 2.28 | 5.21 | 909 | 917 | 520 | 7.0 | 0.0 |
| | **OCR+PAGER** | 0.80 | 12.55 | 5.00 | 2.32 | 5.23 | 914 | 942 | 523 | 7.6 | 0.0 |
| | IMAGES | 0.76 | 12.08 | 0.00 | 7.35 | 4.73 | 4,671 | 1,210 | 473 | 21.6 | 0.0 |
| | IMAGES-AAO | 0.75 | 11.43 | 0.00 | 6.17 | 5.27 | 4,671 | 262 | 526 | 6.7 | 0.0 |
| | **OCR+PAGEN** | *0.70* | 12.66 | 5.00 | 2.38 | 5.28 | 902 | 942 | 523 | 13.4 | 0.0 |
| | OCR+IMAGES | **0.64** | 17.72 | 5.00 | 8.11 | 4.61 | 4,671 | 1,817 | 461 | 18.7 | 0.0 |

similarities (e.g. image quality, document type), or an example of multi-modal inputs assisting with reasoning in general as a version of in-context learning.

### E.4 IAM WITH PYLAIA

As our problem setting is zero-shot and document-level, i.e. no training/fine-tuning data, PYLAIA (and TROCR) poses a problem as it is (i) largely dependent on fine-tuning to achieve reasonable performance on out-of-sample data, and (ii) operates on the line level, rather than the page or document level. We run PYLAIA on an M1 MacBook Pro, CPU only.

We use PYLAIA out-of-the-box with the public `Teklia/pylaia-iam` model, a CNN–BLSTM–CTC recognizer trained on IAM. We enable PYLAIA's lexicon-aware decoding and use the bundled files from the model repository (`tokens.txt`, `lexicon.txt`, `language_model.arpa.gz`). The language model is a 6-gram character LM trained on IAM.

See results in Table 12.

Table 12: Results on IAM. Much of this table is reproduced from Table 1; but new columns with token and cost breakdowns have been added, as well as a new row for PYLAIA. See Tables 26 & 27 for similar reproductions of Tables 2a & 2b respectively.

| | Method | CER ↓ | Cost ($/1k docs) | | | | Tokens (/doc) | | |
| | | | Total | OCR | In | Out | Image | In | Out |
|---|---|---|---|---|---|---|---|---|---|
| — | PYLAIA | 6.51 | 0.00 | 0.00 | 0.00 | 0.00 | 0 | 0 | 0 |
| — | OCR | 3.81 | 2.26 | 2.26 | 0.00 | 0.00 | 0 | 0 | 0 |
| GEMMA | OCR | 2.93 | 2.42 | 2.26 | 0.05 | 0.11 | 0 | 699 | 211 |
| | IMAGES | 2.31 | 0.31 | 0.00 | 0.19 | 0.12 | 2,192 | 547 | 230 |
| | OCR+PAGEN | 2.17 | 2.59 | 2.26 | 0.17 | 0.16 | 945 | 648 | 246 |
| | IMAGES-ALL-AT-ONCE | 1.97 | 0.29 | 0.00 | 0.17 | 0.12 | 2,192 | 262 | 242 |
| | OCR+IMAGES | 1.78 | 2.58 | 2.26 | 0.21 | 0.11 | 2,192 | 821 | 216 |
| | OCR+PAGE1 | 1.36 | 2.50 | 2.26 | 0.11 | 0.12 | 980 | 623 | 243 |
| GPT-4O | OCR | 1.21 | 6.17 | 2.26 | 1.77 | 2.14 | 26 | 682 | 213 |
| | IMAGES-ALL-AT-ONCE | 0.92 | 9.29 | 0.00 | 7.11 | 2.18 | 2,515 | 252 | 218 |
| | IMAGES | 0.92 | 9.81 | 0.00 | 7.63 | 2.17 | 2,515 | 533 | 217 |
| | OCR+PAGEN | 0.87 | 9.04 | 2.26 | 4.52 | 2.26 | 1,095 | 613 | 222 |
| | OCR+PAGE1 | 0.85 | 8.94 | 2.26 | 4.47 | 2.21 | 1,124 | 589 | 221 |
| | OCR+IMAGES | 0.72 | 12.69 | 2.26 | 8.29 | 2.13 | 2,519 | 797 | 213 |
| GEMINI | OCR | 1.57 | 5.21 | 2.26 | 0.87 | 2.08 | 0 | 699 | 207 |
| | IMAGES | 1.16 | 5.56 | 0.00 | 3.43 | 2.13 | 2,192 | 547 | 213 |
| | IMAGES-ALL-AT-ONCE | 0.70 | 5.50 | 0.00 | 3.07 | 2.44 | 2,192 | 262 | 243 |
| | OCR+PAGE1 | *0.65* | 6.63 | 2.26 | 2.00 | 2.37 | 980 | 623 | 236 |
| | OCR+IMAGES | **0.64** | 8.10 | 2.26 | 3.77 | 2.08 | 2,192 | 821 | 207 |
| | OCR+PAGEN | **0.63** | 6.71 | 2.26 | 2.05 | 2.40 | 945 | 648 | 236 |

### E.5 MALVERN-HILLS-5+ WITH TROCR AND DOCOWL2

Experimental details below:

- `Malvern-Hills-5+`: we group consecutive pages (i.e. pages that are continuous sections from the same book of minutes) from `Malvern-Hills` to produce a dataset of 24 documents from 140 images, each with a minimum of 5 pages: 11 documents have 5 pages, 9 have 6, 1 has 7 and 3 have 8 (average: 5.83 pages/doc)

- TROCR: Since our setting is zero-shot, we do not additionally fine-tune TROCR on our datasets, but we do use the `trocr-large-handwritten` model, which is already fine-tuned on handwriting from IAM. As TROCR is intended for line images, we use the kraken command line tool to generate line sub-images of `Malvern-Hills` pages and concatenate results. We run TROCR on an M1 MacBook Pro, CPU only.

- DOCOWL2: we use the DOCOWL2 model hosted on HuggingFace without additional fine-tuning. We use a temperature of zero and a sufficiently high output token limit to ensure no

truncation events. We used 5 rounds of prompt iteration for each mode, testing on a small sample of 5 documents for each. We tested two versions of DOCOWL2: the original, which used full page images as input, and DOCOWL2-LINES, which used cropped line images as input and concatenated them, in a manner identical to TROCR. Our hypothesis was that this might mitigate the early stopping failure mode of DOCOWL2, but it did not work, and only increased inference time significantly (see Table 13). We run DOCOWL2 on an A10 GPU. [5]

See results in Table 13.

Table 13: Results on `Malvern-Hills-5+` for all methods described in the main text, plus OCR+PAGER, TROCR, DOCOWL2 and DOCOWL2-LINES. All failed calls (GEMMA only) were caused by API disconnection without a response.

| | Method | CER ↓ | Cost ($/1k docs) | | | | Tokens (/doc) | | | Time (s/doc) | Fails (/doc) |
|---|---|---|---|---|---|---|---|---|---|---|---|
| | | | Total | OCR | In | Out | Image | In | Out | | |
| — | DOCOWL2-LINES | 93.01 | 0.00 | 0.00 | 0.00 | 0.00 | 0 | 0 | 0 | 2054 | 0.00 |
| | DOCOWL2 | 92.08 | 0.00 | 0.00 | 0.00 | 0.00 | 0 | 0 | 0 | 234 | 0.00 |
| | TROCR | 31.43 | 0.00 | 0.00 | 0.00 | 0.00 | 0 | 0 | 0 | 990 | 0.00 |
| | OCR | 13.96 | 5.83 | 5.83 | 0.00 | 0.00 | 0 | 0 | 0 | — | 0.00 |
| GEMMA | **OCR+PAGEN** | 24.94 | 7.22 | 5.83 | 0.44 | 0.95 | 774 | 2,703 | 1,808 | 126 | 1.12 |
| | **OCR+PAGER** | 23.65 | 7.06 | 5.83 | 0.24 | 0.98 | 774 | 2,703 | 1,967 | 79.4 | 0.58 |
| | **OCR+PAGE1** | 21.35 | 7.03 | 5.83 | 0.24 | 0.95 | 774 | 2,678 | 1,903 | 89.8 | 0.75 |
| | IMAGES-AAO | 17.64 | 1.32 | 0.00 | 0.33 | 0.99 | 4,515 | 262 | 1,980 | 92.4 | 0.75 |
| | IMAGES | 16.00 | 1.43 | 0.00 | 0.41 | 1.01 | 4,515 | 1,411 | 2,025 | 85.4 | 0.58 |
| | OCR | 12.92 | 7.10 | 5.83 | 0.24 | 1.03 | 0 | 3,363 | 2,061 | 44.0 | 0.00 |
| | OCR+IMAGES | 9.84 | 7.40 | 5.83 | 0.57 | 0.99 | 4,515 | 3,678 | 1,988 | 44.8 | 0.00 |
| GPT-4O | IMAGES-AAO | 12.38 | 31.02 | 0.00 | 12.25 | 18.77 | 4,462 | 252 | 1,876 | 80.2 | 0.00 |
| | IMAGES | 10.24 | 33.84 | 0.00 | 14.77 | 19.06 | 4,462 | 1,376 | 1,906 | 90.8 | 0.00 |
| | **OCR+PAGER** | 10.17 | 34.15 | 5.83 | 8.61 | 19.71 | 765 | 2,531 | 1,970 | 46.6 | 0.00 |
| | OCR | 9.42 | 33.70 | 5.83 | 8.35 | 19.52 | 65 | 3,273 | 1,952 | 47.1 | 0.00 |
| | **OCR+PAGE1** | 9.12 | 33.32 | 5.83 | 8.55 | 18.94 | 765 | 2,507 | 1,894 | 51.8 | 0.00 |
| | **OCR+PAGEN** | 7.59 | 33.90 | 5.83 | 8.81 | 19.26 | 765 | 2,531 | 1,921 | 68.2 | 0.00 |
| | OCR+IMAGES | 6.49 | 44.21 | 5.83 | 20.25 | 18.13 | 4,527 | 3,571 | 1,812 | 83.1 | 0.00 |
| GEMINI | OCR | 7.05 | 29.46 | 5.83 | 4.20 | 19.43 | 0 | 3,363 | 1,942 | 24.0 | 0.00 |
| | IMAGES-AAO | 6.14 | 27.12 | 0.00 | 5.97 | 21.15 | 4,515 | 262 | 2,115 | 19.2 | 0.00 |
| | **OCR+PAGER** | 5.99 | 31.22 | 5.83 | 4.35 | 21.04 | 774 | 2,703 | 2,104 | 20.9 | 0.00 |
| | **OCR+PAGEN** | 5.86 | 31.43 | 5.83 | 4.55 | 21.05 | 774 | 2,703 | 2,101 | 31.2 | 0.00 |
| | *OCR+IMAGES* | *5.69* | 35.75 | 5.83 | 10.24 | 19.67 | 4,515 | 3,678 | 1,967 | 27.5 | 0.00 |
| | IMAGES | **5.63** | 27.37 | 0.00 | 7.41 | 19.96 | 4,515 | 1,411 | 1,996 | 30.5 | 0.00 |
| | **OCR+PAGE1** | **5.43** | 31.15 | 5.83 | 4.32 | 21.00 | 774 | 2,678 | 2,099 | 21.4 | 0.00 |

**Discussion.** As with `IAM-5`, even with longer page counts, a PAGEX method remains the best-performing, despite having access to <20% of the raw images per document, along with OCR which has a high error rate on its own. IMAGES alone performs quite well, and is slightly cheaper than OCR+PAGE1 due to the high text density of the `Malvern-Hills` dataset, but it is significantly slower as each image in a multi-page document requires a separate API call.

Unfortunately, our DOCOWL2 and TROCR non-commercial-MLLM baselines perform poorly on this task zero-shot, and are quite slow. TROCR, at least, yields parsable text, but is prone to OCR-like errors such as the misreading of individual words or characters. Conversely, DOCOWL2 is occasionally accurate, but is extremely prone to a number of well-known LLM issues that destroy its overall performance: (i) early stopping (or simply outputting a single token), (ii) egregious hallucination often

[5] DOCOWL2 cannot be run on a CPU-only machine as the `flash-attn` python dependency requires CUDA.

bearing no relation to the actual text, (iii) repetition of words or sentences ad infinitum, (iv) needless addition of code fences or explanatory text (e.g. double quotes, or 'the document reads...'), (v) repeating the prompt, and (vi) ignoring the prompt completely. These behaviors persisted regardless of explicit instructions given against them during prompt iteration.

### E.6 MALVERN-HILLS-10+

We use the same generation method as Malvern-Hills-5+. We obtain 10 documents in total from 115 images; 2 have 10 pages, 2 have 11, 5 have 12, 1 has 13 (average: 11.5 pages/doc).

See results in Table 14.

Table 14: Results on Malvern-Hills-10+ for all methods described in the main text, plus OCR+PAGER. All failed calls (GEMMA only) were a result of incomplete responses (i.e. failure or truncation).

| | Method | CER ↓ | Cost ($/1k docs) | | | | Tokens (/doc) | | | Time (s/doc) | Fails (/doc) |
|---|---|---|---|---|---|---|---|---|---|---|---|
| | | | Total | OCR | In | Out | Image | In | Out | | |
| — | OCR | 12.92 | 11.50 | 11.50 | 0.00 | 0.00 | 0 | 0 | 0 | 0.0 | 0.00 |
| GPT-4O | **OCR+PAGE1** | 21.38 | 56.48 | 11.50 | 12.17 | 32.82 | 765 | 4,677 | 3,281 | 119.1 | 0.40 |
| | **OCR+PAGER** | 17.11 | 56.93 | 11.50 | 12.22 | 33.21 | 765 | 4,701 | 3,321 | 103.9 | 0.30 |
| | **OCR+PAGEN** | 16.37 | 57.09 | 11.50 | 12.58 | 33.01 | 765 | 4,701 | 3,296 | 123.7 | 0.30 |
| | IMAGES-AAO | 11.64 | 59.37 | 0.00 | 23.45 | 35.92 | 8,797 | 252 | 3,592 | 100.8 | 0.20 |
| | IMAGES | 9.90 | 67.13 | 0.00 | 29.12 | 38.01 | 8,797 | 2,714 | 3,800 | 181.1 | 0.00 |
| | OCR | 8.48 | 67.28 | 11.50 | 16.65 | 39.13 | 128 | 6,530 | 3,913 | 93.2 | 0.00 |
| | OCR+IMAGES | 6.44 | 87.86 | 11.50 | 40.11 | 36.26 | 8,925 | 7,116 | 3,625 | 165.4 | 0.00 |
| GEMINI | OCR | 6.88 | 59.29 | 11.50 | 8.40 | 39.39 | 0 | 6,716 | 3,939 | 47.9 | 0.00 |
| | IMAGES-AAO | 6.05 | 54.46 | 0.00 | 11.45 | 43.00 | 8,901 | 262 | 4,300 | 40.9 | 0.00 |
| | **OCR+PAGER** | 5.50 | 61.82 | 11.50 | 7.26 | 43.06 | 774 | 5,033 | 4,306 | 40.1 | 0.00 |
| | **OCR+PAGEN** | 5.41 | 61.77 | 11.50 | 7.62 | 42.65 | 774 | 5,033 | 4,260 | 63.8 | 0.00 |
| | IMAGES | 4.87 | 54.71 | 0.00 | 14.61 | 40.11 | 8,901 | 2,783 | 4,010 | 60.6 | 0.00 |
| | OCR+IMAGES | *4.80* | 71.41 | 11.50 | 20.30 | 39.61 | 8,901 | 7,337 | 3,961 | 54.9 | 0.00 |
| | **OCR+PAGE1** | **4.76** | 61.13 | 11.50 | 7.23 | 42.41 | 774 | 5,008 | 4,240 | 40.6 | 0.00 |

**Discussion.** Even for documents of at least 10 pages in length, OCR+PAGE1 remains the top-performing method, with OCR+IMAGES being the second best, about equivalent in performance, but much slower and more expensive. This suggests that using a single page to improve performance does scale reasonably well, even for quite long documents. This is good news, as it means that, for documents where the average number of text tokens inside each image is less than the average number of tokens produced by tokenizing the image (the case for casia-5, Bentham, IAM and many other real-world datasets — Malvern-Hills is particularly token-dense), then +PAGEX methods will only be more cost and time effective in comparison to full-image methods as document length increases.

### E.7 MALVERN-HILLS METADATA

This section includes information about `Malvern-Hills` at the image level: incidence of challenging OCR features (Table 15) and information about writers (Table 16), time-of-writing (Table 17), and breakdown of pages by primary and secondary content types (Table 18) for each page.

Table 15: `Malvern-Hills` image statistics and prevalence of various OCR challenges. Particularly notable are the reasonably high frequencies of distractor text, tabular data, archaic language and multiple authors.

| | |
|---|---|
| Word count | 219 ± 83 |
| Character count | 1260 ± 473 |
| Includes tabular information | 19.3% |
| Includes margin notes | 15.7% |
| Includes distractor text | 62.1% |
| Non-linear structure | 0.7% |
| Archaic language | 31.4% |
| Poor quality image/damaged paper | 2.1% |
| Handwriting from multiple authors | 18.6% |
| Includes crossed-out text | 23.6% |

Table 16: Unique writers for the `Malvern-Hills` dataset and number of documents containing that writer's handwriting for each. Note that some pages contain multiple hands.

| Author ID | Count |
|---|---|
| 0 | 44 |
| 1 | 23 |
| 2 | 18 |
| 3 | 17 |
| 4 | 16 |
| 5 | 15 |
| 6 | 11 |
| 7 | 6 |
| 8 | 5 |
| 9 | 5 |
| 10 | 2 |
| 11 | 2 |
| 12 | 1 |
| 13 | 1 |

### E.8 MALVERN-HILLS-5+ DOCUMENT TYPE/FEATURE ABLATION

Using primary document types from Table 19 and some choice dataset statistics from Table 15, we ablate document types and features for `Malvern-Hills-5+` in Tables.

Each document of 5 or more pages has a single shared primary type over all individual page images. A document is `tabular`, `archaic` or has `multiiple_hands` if any of its constituent pages has this property.

**Discussion.** Overall we see that the `historical_legal` documents are much more challenging, with $> 3\times$ the proportion of errors compared to `historical_minutes`. We can likely attribute this to (from examination) the ubiquity of archaic language and florid cursive (a product of the much earlier writing dates, $\sim 200$ years prior), which is much harder to transcribe and much more prone to errors in post-processing/correction. While `historical_minutes` documents do include some archaic language, these are generally sporadic instances, rather than comprising the overall style of writing.

Table 17: Years of writing for document pages in `Malvern-Hills`. Some documents were written in one year but copied from a document originally written in another year; where known, this table includes both — for example, some documents use archaic 17th century language but were copied by hand in the 19th century.

| Year | Original | Written |
|---|---|---|
| 1631 | 11 | 0 |
| 1632 | 17 | 17 |
| 1795 | 6 | 6 |
| 1899 | 15 | 15 |
| 1915 | 17 | 17 |
| 1925 | 22 | 22 |
| 1932 | 5 | 5 |
| 1934 | 20 | 20 |
| 1936 | 8 | 8 |
| 1938 | 11 | 11 |
| Unknown | 8 | 8 |

Table 18: Breakdown of primary and secondary page types found in `Malvern-Hills`.

| Primary Type | # Pages | Secondary Types | # Pages |
|---|---|---|---|
| Historical minutes | 98 | — | 72 |
| | | Tabular | 26 |
| Historical legal | 34 | Statute | 17 |
| | | Memoranda roll | 10 |
| | | Case memorandum | 4 |
| | | Memoranda roll, Tabular | 1 |
| | | Case memorandum, Legal letter | 1 |
| | | Legal letter | 1 |
| Historical inventory/schedule | 8 | — | 8 |

Comparing the two document types, we can see that OCR+SINGLE-PAGE methods, and especially OCR+PAGE1, are dominant on the *more challenging archaic document type*, while full-image methods IMAGES, OCR+IMAGES struggle. The reverse is true for the overall *easier* `historical_minutes` document type, where IMAGES, OCR+IMAGES dominate (though OCR+PAGEX methods are still competitive, especially given their comparative image token dearth).

Considering the dominant features of each type of document; it is unsurprising that OCR+PAGE1 is less adept for transcribing documents where a single page includes tabular data, as it is likely that there will not be any on the "seen" page. Conversely, archaic language is likely to be consistent across pages, so single-page extrapolation is more effective in this case.

Overall though, what these results suggest is that OCR+SINGLE-PAGE *methods are most beneficial when the task is more challenging*. We can interpret these results in terms of the tradeoff between prompt complexity and task difficulty. If a task is relatively easier (`_minutes`), the MLLM is less likely to become overwhelmed by a high-complexity or long prompt; it can leverage the additional detail (e.g. all images and OCR) to achieve incremental performance improvement on an almost-solved task. If a task is more challenging, this additional detail/complexity hurts overall performance, and a more optimal balance is achieved with a multi-modal prompt that reduces redundancy — i.e. OCR with a single page.

**Relative importance of document type vs. document features** We can see that document type dominates the differences in performance, — i.e. tables for `tabular`, `multiple_writer`, `non_archaic` generally follow `historical_minutes` (and vice versa for `historical_legal`).

Table 19: For the individual 5+ page *documents* of `Malvern-Hills-5+` (not individual pages/images), the prevalence of tabular data, archaic language and multiple writers for each document. A document is treated as `tabular/archaic/multiple_hands` if any pages have this feature. In general, only one or two pages per document will have tabular information or multiple hands, whereas archaic language will typically be throughout.

| Document type | Num. docs | Num. pages | tabular (%) | archaic (%) | multiple _hands (%) | written_year min \| median \| max |
|---|---|---|---|---|---|---|
| historical _minutes | 17 | 98 | 88.2 | 41.2 | 82.4 | 1899 \| 1925 \| 1938 |
| historical _legal | 6 | 34 | 16.7 | 100 | 33.3 | 1632 \| 1713 \| 1884 |

Table 20: Performance for `historical_minutes`-typed documents only from `Malvern-Hills-5+`. All methods use GEMINI-2.5-PRO as the MLLM.

| Method | CER (%) |
|---|---|
| Azure OCR only | 11.8 |
| OCR | 5.3 |
| IMAGES-AAO | 4.55 |
| OCR+PAGER | 3.88 |
| OCR+PAGEN | 3.83 |
| OCR+PAGE1 | 3.8 |
| IMAGES | 3.16 |
| OCR+IMAGES | 3.11 |

E.9 NEW COLUMNS FOR EXISTING TABLES

Our new experiments include additional data tracking per-stage tokens and costs, as well as inference times and failure rates. As we did not record failure rates or inference times when originally performing experiments, we cannot recover these, but the new experimental data in the rest of this section provides examples for (versions of) the `IAM`, `Malvern-Hills` and `casia-5` datasets.

We can, however, compute per-stage tokens and costs for our existing results in the main text. Tables 12, 26 & 27 correspond to Tables 1, 2a & 2b in the main text, and reproduce performance scores, but add new cost breakdown columns.

Table 21: Performance for `historical_legal`-typed documents only from `Malvern-Hills-5+`. All methods use GEMINI-2.5-PRO as the MLLM.

| Method | CER (%) |
|---|---|
| Azure OCR only | 20.81 |
| OCR+IMAGES | 13.56 |
| IMAGES | 13.17 |
| OCR | 12.53 |
| OCR+PAGER | 12.36 |
| OCR+PAGEN | 12.03 |
| IMAGES-AAO | 11.11 |
| OCR+PAGE1 | 10.36 |

Table 22: Performance for `historical_inventory/schedule`-typed documents only from `Malvern-Hills-5+`. All methods use GEMINI-2.5-PRO as the MLLM.

| Method | CER (%) |
|---|---|
| Azure OCR only | 9.58 |
| OCR | 4.06 |
| OCR+PAGER | 3.56 |
| OCR+PAGE1 | 3.46 |
| OCR+PAGEN | 3.41 |
| IMAGES-AAO | 3.27 |
| IMAGES | 2.42 |
| OCR+IMAGES | 2.34 |

Table 23: Performance for documents from `Malvern-Hills-5+` split by presence of tabular content. A document is `tabular` if any constituent page includes tabular layout. All methods use GEMINI-2.5-PRO as the MLLM.

(a) `tabular` documents

| Method | CER (%) |
|---|---|
| Azure OCR only | 12.18 |
| OCR | 5.56 |
| IMAGES-AAO | 4.56 |
| OCR+PAGER | 4.00 |
| OCR+PAGE1 | 3.98 |
| OCR+PAGEN | 3.97 |
| IMAGES | 3.28 |
| OCR+IMAGES | 3.20 |

(b) `non_tabular` documents

| Method | CER (%) |
|---|---|
| Azure OCR only | 17.53 |
| OCR+IMAGES | 10.68 |
| IMAGES | 10.35 |
| OCR | 10.05 |
| OCR+PAGER | 9.96 |
| OCR+PAGEN | 9.63 |
| IMAGES-AAO | 9.28 |
| OCR+PAGE1 | 8.31 |

Table 24: Performance for documents from `Malvern-Hills-5+` split by presence of archaic language. A document is `archaic` if any constituent page is marked as such. All methods use GEMINI-2.5-PRO as the MLLM.

(a) `archaic` documents

| Method | CER (%) |
|---|---|
| Azure OCR only | 16.50 |
| OCR | 8.61 |
| OCR+IMAGES | 7.96 |
| IMAGES | 7.86 |
| OCR+PAGER | 7.74 |
| IMAGES-AAO | 7.71 |
| OCR+PAGEN | 7.58 |
| OCR+PAGE1 | 6.95 |

(b) `non_archaic` documents

| Method | CER (%) |
|---|---|
| Azure OCR only | 10.96 |
| OCR | 5.22 |
| IMAGES-AAO | 4.28 |
| OCR+PAGER | 3.92 |
| OCR+PAGEN | 3.82 |
| OCR+PAGE1 | 3.63 |
| OCR+IMAGES | 3.01 |
| IMAGES | 3.00 |

Table 25: Performance for documents from `Malvern-Hills-5+` split by whether they contain handwriting from multiple authors. A document has `multiple_hands` if any page is written by a different author. All methods use GEMINI-2.5-PRO as the MLLM.

(a) `multiple_hands` documents

| Method | CER (%) |
|---|---|
| Azure OCR only | 12.35 |
| OCR | 5.73 |
| IMAGES-AAO | 5.12 |
| OCR+PAGER | 4.79 |
| OCR+PAGEN | 4.76 |
| OCR+PAGE1 | 4.15 |
| IMAGES | 3.85 |
| OCR+IMAGES | 3.55 |

(b) `single_hand` documents

| Method | CER (%) |
|---|---|
| Azure OCR only | 17.19 |
| OCR+IMAGES | 9.97 |
| OCR | 9.70 |
| IMAGES | 9.20 |
| OCR+PAGER | 8.38 |
| IMAGES-AAO | 8.17 |
| OCR+PAGEN | 8.07 |
| OCR+PAGE1 | 7.98 |

Table 26: Results on `Malvern-Hills`. Much of this table is reproduced from Table 2a; but new columns with token and cost breakdowns have been added.

| | Method | CER ↓ | Cost ($/1k docs) | | | | Tokens (/doc) | | |
|---|---|---|---|---|---|---|---|---|---|
| | | | Total | OCR | In | Out | Image | In | Out |
| — | OCR | 14.41 | 2.30 | 2.30 | 0.00 | 0.00 | 0 | 0 | 0 |
| GEMMA | IMAGES | 27.19 | 0.74 | 0.00 | 0.16 | 0.58 | 1,791 | 556 | 1,152 |
| | IMAGES-ALL-AT-ONCE | 15.21 | 0.66 | 0.00 | 0.14 | 0.52 | 1,791 | 262 | 1,037 |
| | OCR | 13.52 | 2.80 | 2.30 | 0.09 | 0.41 | 0 | 1,331 | 818 |
| | **OCR+PAGE1** | 12.55 | 2.89 | 2.30 | 0.14 | 0.44 | 781 | 1,285 | 888 |
| | **OCR+PAGEN** | 11.22 | 3.03 | 2.30 | 0.25 | 0.48 | 781 | 1,317 | 879 |
| | OCR+IMAGES | 10.54 | 2.92 | 2.30 | 0.23 | 0.40 | 1,791 | 1,455 | 791 |
| GPT-4O | IMAGES-ALL-AT-ONCE | 11.35 | 12.97 | 0.00 | 5.31 | 7.66 | 1,774 | 252 | 765 |
| | IMAGES | 11.24 | 13.46 | 0.00 | 5.86 | 7.60 | 1,774 | 542 | 760 |
| | OCR | 10.60 | 13.32 | 2.30 | 3.31 | 7.71 | 25 | 1,298 | 771 |
| | **OCR+PAGE1** | 8.92 | 15.00 | 2.30 | 5.16 | 7.53 | 774 | 1,203 | 753 |
| | **OCR+PAGEN** | 8.05 | 15.30 | 2.30 | 5.35 | 7.65 | 774 | 1,235 | 761 |
| | OCR+IMAGES | 7.11 | 17.54 | 2.30 | 8.04 | 7.20 | 1,799 | 1,415 | 720 |
| GEMINI | IMAGES-ALL-AT-ONCE | 8.20 | 11.01 | 0.00 | 2.57 | 8.45 | 1,791 | 262 | 844 |
| | OCR | 7.47 | 11.67 | 2.30 | 1.66 | 7.70 | 0 | 1,331 | 770 |
| | **OCR+PAGEN** | 6.54 | 14.56 | 2.30 | 2.73 | 9.53 | 781 | 1,317 | 948 |
| | OCR+IMAGES | 6.46 | 14.18 | 2.30 | 4.06 | 7.82 | 1,791 | 1,455 | 782 |
| | *IMAGES* | *6.42* | 10.88 | 0.00 | 2.93 | 7.95 | 1,791 | 556 | 794 |
| | **OCR+PAGE1** | **5.83** | 13.24 | 2.30 | 2.58 | 8.36 | 781 | 1,285 | 835 |

Table 27: Results on `Bentham`. Much of this table is reproduced from Table 2b; but new columns with token and cost breakdowns have been added.

| | Method | CER ↓ | Cost ($/1k docs) | | | | Tokens (/doc) | | |
| | | | Total | OCR | In | Out | Image | In | Out |
|---|---|---|---|---|---|---|---|---|---|
| — | OCR | 11.18 | 2.63 | 2.63 | 0.00 | 0.00 | 0 | 0 | 0 |
| **GEMMA** | IMAGES-ALL-AT-ONCE | 25.06 | 0.55 | 0.00 | 0.18 | 0.37 | 2,287 | 262 | 747 |
| | IMAGES | 15.60 | 0.70 | 0.00 | 0.20 | 0.50 | 2,287 | 635 | 990 |
| | **OCR+PAGEN** | 11.02 | 3.34 | 2.63 | 0.25 | 0.47 | 871 | 1,269 | 851 |
| | **OCR+PAGE1** | 10.89 | 3.20 | 2.63 | 0.15 | 0.42 | 871 | 1,244 | 847 |
| | OCR | 10.75 | 3.11 | 2.63 | 0.10 | 0.39 | 0 | 1,367 | 781 |
| | OCR+IMAGES | 9.98 | 3.28 | 2.63 | 0.27 | 0.39 | 2,287 | 1,509 | 781 |
| **GPT-4O** | **OCR+PAGE1** | 10.95 | 16.29 | 2.63 | 5.97 | 7.69 | 1,105 | 1,189 | 769 |
| | IMAGES-ALL-AT-ONCE | 10.18 | 15.80 | 0.00 | 8.15 | 7.65 | 2,902 | 252 | 764 |
| | OCR | 9.97 | 13.63 | 2.63 | 3.42 | 7.58 | 29 | 1,338 | 758 |
| | IMAGES | 9.87 | 16.58 | 0.00 | 8.88 | 7.69 | 2,902 | 619 | 769 |
| | OCR+IMAGES | 9.35 | 20.93 | 2.63 | 11.01 | 7.30 | 2,931 | 1,472 | 729 |
| | **OCR+PAGEN** | 8.87 | 16.37 | 2.63 | 6.13 | 7.61 | 1,105 | 1,213 | 757 |
| **GEMINI** | IMAGES-ALL-AT-ONCE | 9.88 | 11.70 | 0.00 | 3.19 | 8.51 | 2,287 | 262 | 851 |
| | IMAGES | 9.74 | 11.64 | 0.00 | 3.65 | 7.99 | 2,287 | 635 | 798 |
| | OCR | 9.54 | 12.03 | 2.63 | 1.71 | 7.70 | 0 | 1,367 | 769 |
| | OCR+IMAGES | 8.67 | 15.32 | 2.63 | 4.75 | 7.95 | 2,287 | 1,509 | 794 |
| | *OCR+PAGE1* | *8.56* | 13.55 | 2.63 | 2.64 | 8.28 | 871 | 1,244 | 827 |
| | **OCR+PAGEN** | **8.48** | 13.70 | 2.63 | 2.77 | 8.30 | 871 | 1,269 | 825 |