# OpenReview forum: "Judge a Book by its Cover: Investigating Multi-Modal LLMs for Multi-Page Handwritten Document Transcription"
_ICLR.cc/2026/Conference — Submitted to ICLR 2026_

### Official Review · Reviewer_tXKd · 2025-10-31

**Soundness:** 3
**Presentation:** 3
**Contribution:** 2
**Rating:** 6
**Confidence:** 4

**Summary:**

This work introduces a benchmark framework for transcribing multi-page handwritten documents through a two-step process: initial transcription using OCR engines, followed by refinement with multimodal large language models (MLLMs). While OCR provides a preliminary draft, zero-shot prompting enables MLLMs to detect and correct errors at both character and semantic levels. The study proposes two architectural variants—OCR+PAGE1 and OCR+PAGEN—which combine OCR output with a selected document image to guide the correction process. The key distinction lies in the choice of image used for prompting. By incorporating visual context, the system is hypothesized to learn and generalize transcription errors across pages. Evaluation is conducted across multiple benchmark tasks, including the proposed methods, and is supported by the release of a new dataset: Malvern-Hills.

**Strengths:**

Originality:
Tackles the challenging problem of multi-page handwritten document transcription, emphasizing semantic continuity across pages.
Introduces a zero-shot prompting strategy for MLLMs, which adds a novel layer of semantic error correction beyond character-level transcription.
Proposes two architectural variants, OCR+PAGE1 and OCR+PAGEN, that leverage partial visual context to guide transcription refinement.

Quality:
Benchmarks include single-page and multi-page setups, offering a comprehensive view of system performance under varied conditions.
The experimental design, comparing multiple configurations across OCR and MLLM modules, supports the architectural choices and hypotheses, demonstrating empirical rigor.

Clarity:
The paper is clearly written and well-structured, with hypotheses and contributions articulated in an accessible manner.
The benchmark structure and dataset contributions are presented in a way that is accessible and reproducible.

Significance:
Contributes a new dataset (Malvern-Hills) and multi-page extensions of existing datasets, which will be valuable resources for the research community.
Establishes a benchmark framework that can guide future work in multi-page handwriting transcription.

**Weaknesses:**

The proposed approach relies heavily on off-the-shelf commercial components (OCR engines and MLLMs), which limits the novelty from a methodological standpoint. While the architectural design and prompting strategy are thoughtfully constructed, the paper would benefit from deeper algorithmic innovation or theoretical grounding.

The multi-page dataset construction primarily involves grouping existing pages, which may not fully reflect the diversity and complexity of real-world multi-page documents. In practice, documents vary significantly in writer identity, layout structure, and semantic content. The paper would be strengthened by a dedicated analysis of how performance varies across different document types or heterogeneous sources (e.g., letters vs. forms, single-writer vs. multi-writer).

The Malvern-Hills dataset, while a valuable addition, is somehow underutilized in the analysis. Given its novelty, the authors should consider to provide: ablation studies isolating the impact depending on variations in the types of documents. Metadata analysis (e.g., writer diversity, layout variation) to demonstrate its relevance and richness compared to existing datasets.

**Questions:**

Please take into account the comments outlined above. Overall, the paper presents several positive contributions. However, for a more definitive assessment, I would like to see stronger evidence of scientific and methodological innovation beyond the integration of existing components. Additionally, the proposed dataset is a valuable asset, and its internal heterogeneity offers an opportunity for deeper evaluation and discussion. A more comprehensive analysis of its characteristics and impact would significantly enhance the paper’s contribution.

---

> ### Author Response · Authors · 2025-11-25
> **Response to reviewer tXKd (1/2)**
>
> We thank the reviewer for their detailed engagement with our work, their positive feedback, and for their patience during the review process. We have run comprehensive additional experiments, following the suggestions of several reviewers, which we believe greatly improve on our existing work. We have summarised them in an official comment above as a response to all reviewers. We would be grateful if the reviewer would take into consideration these new results, which we refer to in this response.
>
> We attempt to address the reviewer’s concerns below.
>
> ---
>
> > The proposed approach relies heavily on off-the-shelf commercial components (OCR engines and MLLMs), which limits the novelty from a methodological standpoint. While the architectural design and prompting strategy are thoughtfully constructed, the paper would benefit from deeper algorithmic innovation or theoretical grounding.
>
> We thank the reviewer for raising this point. Our additional experiments, specifically Tables R4 and R5, include evaluation of three additional non-commercial models frequently used for handwriting text recognition; PyLaia, TrOCR and DocOwl2.
>
> Unfortunately, we find that these models all perform quite poorly without any additional fine-tuning, and our problem setting is zero-shot, to reflect real-world paucity of labelled data. Simply put, it appears that open models cannot compete with commercial tools for this task, and as we are unable to modify or recreate these tools ourselves, opportunity for theoretical innovation beyond prompt engineering is somewhat limited.
>
> Nevertheless, we believe that our empirical results remain valuable. Ours is a broad, novel and comprehensive study. By focusing on a specific, more challenging instance of OCR (handwritten, multi-page documents), and on multi-modal MLLM prompting, we believe our work is of interest to both the OCR/document processing/vision community, and to the MLLM research community.
>
> ---
>
> > The Malvern-Hills dataset, while a valuable addition, is somehow underutilized in the analysis. Given its novelty, the authors should consider to provide: ablation studies isolating the impact depending on variations in the types of documents. Metadata analysis (e.g., writer diversity, layout variation) to demonstrate its relevance and richness compared to existing datasets.
>
> We thank the reviewer for raising this excellent point. We have updated our paper to include detailed metadata about the `Malvern-Hills` dataset, including frequency of features which are challenging for OCR such as tabular data, multiple authors, distractor text (text present in the image that is not part of the given document page), structure, and issues with image or paper quality.
>
> The suggestion to include ablation studies considering specific document types or features in isolation is a good one.
>
> In a sense, our existing experiments over multiple datasets can be seen as a step in this direction, as each dataset has unique features that affect the performance of transcription methods. For example, `IAM` is a relatively easy task due to its low-noise, processed images, the lack of multiple hands, distractors or tabular information, and the modern and typically legible handwriting/language. `CASIA-5` is similar in most of these ways to `IAM`, but is much more challenging as it is in Chinese. `Bentham` is in English but is quite challenging, as it consists of a difficult cursive script and noisy images, but the documents are all of a similar type: personal notes, and all in one hand. `Malvern-Hills` is perhaps the most varied, with noisy images, language and documents spread out over hundreds of years, text written in a variety of different hands, with document types including meeting minutes, legal letters and notes and tabular information.
>
> We will update the final version of the paper to detail more clearly the differences between different datasets, and discuss how sources affect transcription.
>
> Regarding a specific ablation study on different traits of `Malvern-Hills`; we have not yet had time to do this during the review period due to the extent of our other experiments, but as we now have `Malvern-Hills` document metadata, we will endeavour to provide such additional results in the remainder of the review period.

---

> > ### Author Response · Authors · 2025-11-25
> > **Response to reviewer tXKd (2/2)**
> >
> > > The multi-page dataset construction primarily involves grouping existing pages, which may not fully reflect the diversity and complexity of real-world multi-page documents. In practice, documents vary significantly in writer identity, layout structure, and semantic content. The paper would be strengthened by a dedicated analysis of how performance varies across different document types or heterogeneous sources (e.g., letters vs. forms, single-writer vs. multi-writer).
> >
> > As mentioned above, we will endeavour to provide a dedicated analysis of the impact of specific document types/features for the `Malvern-Hills` dataset, which contains many documents with multiple hands and types.
> >
> > More broadly, the discussion about what constitutes a real-world multi-page document is an interesting one. In this paper we assume that a multi-page document possesses at least some degree of similarity between pages — this may be in handwriting, semantic content, image quality/style, document type or other. It could be argued that a document with no such similarities is not really a single document at all, but a collection of separate ones. Indeed, the types of multi-page documents one is likely to encounter in the real world, such as letters, notebooks, minutes, legal documents, etc., are likely to possess several such similarities across pages — `Bentham` and `Malvern-Hills` are examples of this.
> >
> > As a first step towards such ablations as the reviewer describes, we detail in Table R3 and the updated paper an ablation on a 5-page, writer-randomized version of `IAM`, `IAM-5-Random`, to investigate the effect on transcription accuracy when neither semantic content *nor* handwriting are consistent. As expected — as our methods are intended to exploit such cross-page similarities — performance is weaker, and the strongest-performing method is `OCR+IMAGES`. Nevertheless, `OCR+PAGE-N` still outperforms MLLM methods that access to all images, despite having access to only 20% of them, and lower-score OCR output.
> >
> > ---
> >
> > We thank the reviewer again for their engagement with our work, detailed feedback, and patience during the review process. We hope we have addressed some of the reviewer's concerns, and, in light of our response and comprehensive additional experiments, we ask the reviewer to kindly consider raising their score.
> >
> > We are happy to answer any further questions.

---

> ### Author Response · Authors · 2025-11-28
> **Additional results: `Malvern-Hills` document type ablation**
>
> As suggested by the reviewer, we include ablations on `Malvern-Hills-5+` by document type, to supplement Tables R7-9.
>
> ---
>
> **Table R10:** For the individual 5+ page *documents* of `Malvern-Hills-5+` (not individual pages/images), the prevalence of tabular data, archaic language and multiple writers for each document. A document is treated as `tabular`/`archaic`/`multiple_hands` if any pages have this feature. In general, only one or two pages per document will have tabular information or multiple hands, whereas archaic language will typically be *throughout*. We omit the one remaining document type, for which there is only a single eight-page document, and the written year is unknown.
>
> | Document type        | Num. docs | Num. pages | `tabular` (%) | `archaic` (%) | `multiple_hands` (%) | `written_year` (min \| median \| max) |
> | :------------------- | -----------: | ------------: | ---------------: | ---------------: | ----------------------------: | :-------------------------------------- |
> | `historical_minutes` |           17 |            98 |             88.2 |             41.2 |                          82.4 | 1899 \| 1925 \| 1938                    |
> | `historical_legal`   |            6 |            34 |             16.7 |              100 |                          33.3 | 1632 \| 1713 \| 1884                    |
>
> ---
>
> **Table R11:** Performance for **`historical_minutes`**-typed documents only from `Malvern-Hills-5+`. All methods use `gemini-2.5-pro` as the MLLM.
>
> |                | CER (%) |
> | :------------- | ------: |
> | Azure OCR only |    11.8 |
> | `OCR`          |     5.3 |
> | `IMAGES-AAO`   |    4.55 |
> | `OCR+PAGE-R`   |    3.88 |
> | `OCR+PAGE-N`   |    3.83 |
> | `OCR+PAGE-1`   |     3.8 |
> | `IMAGES`       |    3.16 |
> | `OCR+IMAGES`   |    3.11 |
>
> ---
>
> **Table R12:** Performance for **`historical_legal`**-typed documents only from `Malvern-Hills-5+`. All methods use `gemini-2.5-pro` as the MLLM.
>
> |                | CER (%) |
> | :------------- | ------: |
> | Azure OCR only |   20.81 |
> | `OCR+IMAGES`   |   13.56 |
> | `IMAGES`       |   13.17 |
> | `OCR`          |   12.53 |
> | `OCR+PAGE-R`   |   12.36 |
> | `OCR+PAGE-N`   |   12.03 |
> | `IMAGES-AAO`   |   11.11 |
> | `OCR+PAGE-1`   |   10.36 |
>
> ---
>
> ### Discussion
> Overall we see that the `historical_legal` documents are much more challenging, with $>3\times$ the proportion of errors compared to `historical_minutes`. We can likely attribute this to (from examination) the ubiquity of archaic language and florid cursive (a product of the much earlier writing dates, ~200 years prior), which is much harder to transcribe and much more prone to errors in post-processing/correction. While `historical_minutes` documents do include some archaic language, these are generally sporadic instances, rather than comprising the overall style of writing.
>
> Comparing the two document types, we can see that OCR+single-page methods, and especially `+PAGE-1`, are dominant on the *more challenging archaic document type*, while full-image methods `IMAGES`, `OCR+IMAGES` struggles. The reverse is true for  the overall *easier* `historical_minutes` document type, where `IMAGES`, `OCR+IMAGES` dominate (though `+PAGE-X` methods are still competitive, especially given their comparative image token dearth).
>
> Considering the dominant features of each type of document; it is unsurprising that `OCR+PAGE-1` is less adept for transcribing documents where a single page includes tabular data, as it is likely that there will not be any on the 'seen' page. Conversely, archaic language is likely to be consistent across pages, so single-page extrapolation is more effective in this case.
>
> Overall though, what these results suggest is that *OCR+single-page methods are most beneficial when the task is more challenging*. We can interpret these results in terms of the tradeoff between prompt complexity and task difficulty. If a task is relatively easier (`_minutes`), the MLLM is less likely to become overwhelmed by a high-complexity or long prompt; it can leverage the additional detail (e.g. all images and OCR) to achieve incremental performance improvement on an almost-solved task. If a task is more challenging, this additional detail/complexity hurts overall performance, and a more optimal balance is achieved with a multi-modal prompt that reduces redundancy — i.e. OCR with a single page.
>
> N.B. Individual tables by `tabular`/`non_tabular`, `multiple_hands`/`single_hand`, `archaic`/`non_archaic` are provided in the updated paper, but here we focus only on the document type, as this is what dominates the differences in performance — i.e. tables for `tabular`, `multiple_writer`, `non_archaic` generally follow `historical_minutes` (and vice versa for `historical_legal`).

---

### Official Review · Reviewer_sUZQ · 2025-10-31

**Soundness:** 4
**Presentation:** 3
**Contribution:** 3
**Rating:** 8
**Confidence:** 5

**Summary:**

This paper introduces new methods that use MLLM to improve the OCR transcription of handwritten documents. The authors compare several inputs for the MLLM, including the use of OCR transcriptions and a combination of images with their corresponding OCR transcriptions. The proposed approach improves the results of existing OCR systems, which often perform poorly on handwritten documents when not fine-tuned. The approach also exploits the redundancy of handwriting styles across pages to improve the overall transcription quality while reducing post-processing costs. The authors demonstrate the effectiveness of their approach on three datasets, including one specially designed for evaluation purposes.

**Strengths:**

The paper is well written, clearly structured and easy to follow, except for the explanation of the methods on page 5, which may slightly confuse the reader. The arguments are well illustrated and justified. In addition, it is particularly important to address costs when working with MLLMs, and this aspect is discussed throughout the paper. While not really emphasised, the evaluation of the different types of errors using an LLM is also very interesting and is rarely presented in papers. It is crucial to understand which errors persist and are added by the model in order to identify how they can be solved.
Furthermore, the proposed approach is easily reusable, as it is straightforward to implement. It seems that the prompts are available in the GitHub repository, which enhances the reproducibility of the approach. The paper is sufficiently clear to allow readers to reproduce the results and adapt the approach to their own data.

**Weaknesses:**

The pages selected as input for the post-OCR correction model are either the first page or a page chosen by an intermediate LLM. It would have been interesting to compare these approaches with a random page selection. This would be a compromise, avoiding the systematic use of the first page, which may only contain a title, while simplifying the processing pipeline by removing the need to use an LLM only for page selection.
Furthermore, the paper frequently refers to “multi-page” processing. However, most of the documents used in the experiments contain only two pages. While this demonstrates some improvement, the use of the pageN to select a single page out of two seems rather complex. Evaluating the proposed methods on longer documents (e.g., at least five or ten pages) would provide a more realistic evaluation of their effectiveness

**Questions:**

The costs of the models were discussed throughout the paper. It would also have been interesting to analyse inference times, as these can sometimes be critical when processing long documents or large amount of documents.

---

> ### Author Response · Authors · 2025-11-25
> **Response to reviewer sUZQ**
>
> We thank the reviewer for their detailed engagement with our work, for their patience during the review process, and for their effusive positive feedback.
>
> We have run comprehensive additional experiments, following the suggestions of several reviewers, which we believe greatly improves on our existing work. We have summarised them in an official comment above as a response to all reviewers. We would be grateful if the reviewer would take into consideration these new results, which we refer to in this response.
>
> We attempt to address the reviewer’s concerns below.
>
> ---
>
> > It would have been interesting to compare these approaches with a random page selection. This would be a compromise, avoiding the systematic use of the first page, which may only contain a title, while simplifying the processing pipeline by removing the need to use an LLM only for page selection.
>
> We thank the reviewer for this very interesting suggestion. We have run extensive new experiments, which include a new method, `OCR+PAGE-R`, which, as the reviewer describes, performs `OCR+PAGE-N` (using the same post-processing prompt) with a page chosen at random.
>
> It is interesting to see in our new experiments that `+PAGE-R` almost always underperforms compared to `+PAGE-N`, demonstrating that the upstream MLLM is indeed typically choosing more useful/representative images. For some tasks, `+PAGE-1` still outperforms both `+PAGE-R` and `+PAGE-N`; we can likely attribute this to a combination of increased prompt complexity for `+PAGE-R`/`-N` relative to `+PAGE-1`, and uniform usefulness across pages (see Appendix C2 for a discussion of this).
>
> ---
>
> > Most of the documents used in the experiments contain only two pages ... Evaluating the proposed methods on longer documents (e.g., at least five or ten pages) would provide a more realistic evaluation of their effectiveness
>
> We thank the reviewer for raising this point, which we have addressed with additional experiments. We run all methods on one new (Chinese handwriting) dataset, `CASIA-5`, with five pages per document, and three re-workings of our existing datasets — `IAM-5`, `Malvern-Hills-5+` and `Malvern-Hills-10+` — with average page counts of 5, 5.8, and 11.5 respectively. We find that single-page+OCR continue to be the top-performing methods, or comparable with `OCR+IMAGES`, for all four datasets, suggesting that these methods scale well. Please see our official comments and the updated Appendix of the paper for full details.
>
> ---
>
> > It would also have been interesting to analyse inference times, as these can sometimes be critical when processing long documents or large amount of documents.
>
> We thank the reviewer for this excellent suggestion; our experiments now include inference times; see official comments and the Appendix of the updated paper for further details. This analysis has shown that in addition to attaining strong performance and at a reasonable cost, OCR + single-page methods methods are also much *faster* than page-by-page methods.
>
> ---
>
> > The paper is well written, clearly structured and easy to follow, except for the explanation of the methods on page 5, which may slightly confuse the reader.
>
> We are grateful to the reviewer for pointing this out. We will ensure that the explanations of methods are clearer in the final version.
>
> ---
>
> We thank the reviewer again for their engagement with our work, detailed feedback, and patience during the review process. We hope we have addressed the reviewer's concerns, and, in light of our response and comprehensive additional experiments, we ask the reviewer to kindly consider raising their score.
>
> We are happy to answer any further questions.

---

### Official Review · Reviewer_5mTv · 2025-10-31

**Soundness:** 3
**Presentation:** 3
**Contribution:** 2
**Rating:** 2
**Confidence:** 5

**Summary:**

This paper evaluates using multi-modal LLMs for handwritten document transcription. Specifically, the paper focuses on the problem of multi-page OCR. In this context, it evaluates several alternatives for presenting the LLM with image + text information - OCR + images, OCR + page1, and OCR + pageN, where the difference in the image modality is having the specific page to be transcribed, the first page, and a single page chosen by an upstream LLM, respectively.

Results over three datasets show the proposed OCR + page1 and OCR + pageN to be among the top performing methods, and generally comparable to OCR + images, thus demonstrating that information from a single page can be extrapolated to improve OCR overall, at a level similar to having access to the full set of images, thereby helping to reduce total transcription cost.

**Strengths:**

Thorough evaluation and discussion.  In particular, the proposed method is tested on three separate datasets, including one dataset constructed by the authors.

**Weaknesses:**

Technical novelty/contribution is minor, as the method is a small modification to the baseline of OCR + images, the problem is somewhat niche (minimizing cost in multi-page handwriting OCR), and the improvement/benefit is somewhat small. Paper is probably better suited to a more text/OCR-specific conference.

In addition, the multi-page documents are generally short, the majority being 2 pages, and only one dataset having a small number of 4 page documents, so both the challenge and benefit of using one page versus the whole document seems limited.  It would potentially be more interesting if longer documents were used (eg 10+ pages).  One potential question would be how performance scales with page length.

**Questions:**

see weaknesses

---

> ### Author Response · Authors · 2025-11-25
> **Response to reviewer 5mTv (1/2)**
>
> We thank the reviewer for their detailed engagement with our work, and for their patience during the review process. We have run comprehensive additional experiments, following the suggestions of several reviewers, which we believe greatly improve on our existing work. We have summarised them in an official comment above as a response to all reviewers. We would be grateful if the reviewer would take into consideration these new results, which we refer to in this response.
>
> We attempt to address the reviewer’s concerns below.
>
> ---
>
> > It would potentially be more interesting if longer documents were used (eg 10+ pages). One potential question would be how performance scales with page length.
>
> We thank the reviewer for this point. We have run additional experiments on one new dataset (`CASIA-5`, Chinese handwritten documents) and three re-workings of existing datasets (`Malvern-Hills-5+`, `Malvern-Hills-10+` `IAM-5+`) to explore performance on datasets of documents with 5-13 pages. We kindly ask the reviewer to see our above official comments and the updated Appendix in the paper for full details, but we summarize below.
>
> In all cases we find that either the best-performing model is one of our single-page methods, or that single page methods perform similarly well to full `OCR+IMAGES`, at lower cost and with faster inference time. Furthermore, as our methods use only a single image regardless of document length, the cost savings only increase with longer documents.
>
> ---
>
> > The problem is somewhat niche (minimizing cost in multi-page handwriting OCR)
>
> We respectfully disagree. OCR is a major area, with broad applications across industry, research and the social sciences, and the use of MLLMs in this area is of interest not only to researchers involved with OCR/document processing, but to computer vision and MLLM researchers in general.
>
> Furthermore, the specific cases of handwriting and of multi-page documents do not narrow the scope of the paper but actually widen it. Handwriting OCR, especially of photos of historical documents (as with `Bentham`, `Malvern-Hills`) is far more challenging than OCR in general; any advances in handwriting OCR are applicable to OCR more broadly.
>
> In a similar vein, focusing on multi-page documents does not narrow the scope but widens it to look at a more challenging, more general problem. We tackle the problem of transcription/OCR *at point of entry for practitioners*, i.e. a practitioner has a stack of unseparated, noisy page images, not a series of perfectly cropped and post-processed line images, the use case that most OCR research focuses on.
>
> Lastly, our contribution goes beyond minimizing cost, as discussed below.
>
> ---
>
> > Technical novelty/contribution is minor, as the method is a small modification to the baseline of OCR + images ... and the improvement/benefit is somewhat small.
>
> We respectfully disagree that the contribution is minor. Firstly, our methods show consistent improvement over several different datasets at varying document lengths. The reviewer notes that this performance improvement is minor, which is fair, but our intention with this paper is not primarily to report SOTA results.
>
> Our OCR + single-page methods are not a small and needlessly complex additional tweak on an existing method; instead, they are actually a *simplification* of OCR+IMAGES; cheaper, easier, faster, more scalable. The significance of our findings is not in the size of the performance gap, but the fact that OCR+PAGE-1 and +PAGE-N *perform on par with* OCR+IMAGES *at all*. It is quite remarkable that, given OCR output which scores poorly on its own, and only a fraction of document images (in the case of `Malvern-Hills-10+`, less than 10%), MLLMs are able to produce *more accurate transcriptions* than they are when given access to all images. Beyond any practical and economic utility, this is also an interesting example of MLLM's ability to extrapolate, and an interesting example of how MLLMs can leverage limited input to learn challenging tasks in-context.
>
> Furthermore, we would like to respectfully invite the reviewer to acknowledge the utility of our work as a reference for the community, and for an understudied problem. We do not only introduce new methods, we provide comprehensive experiments on a suite of transcription methods and prompting strategies that, to our knowledge, have not been directly compared in previous works. We provide a benchmarking framework for multi-page document transcription, and, most significantly, *introduce a brand new labeled dataset*. **`Malvern-Hills`** is a novel and challenging transcription dataset, a useful resource for the community. We believe these are all nontrivial and meaningful contributions to an under-explored area.

---

> > ### Author Response · Authors · 2025-11-25
> > **Response to reviewer 5mTv (2/2)**
> >
> > We thank the reviewer again for their engagement with our work, detailed feedback, and patience during the review process. We hope we have addressed the reviewer's concerns, and, in light of our response and comprehensive additional experiments, we ask the reviewer to kindly consider raising their score.
> >
> > We are happy to answer any further questions.

---

### Official Review · Reviewer_71my · 2025-10-31

**Soundness:** 3
**Presentation:** 3
**Contribution:** 3
**Rating:** 4
**Confidence:** 4

**Summary:**

The authors propose an interesting idea, suggesting that one page image can provide enough visual context to correct OCR across the document. From this claim, the paper studies zero-shot transcription of multi-page handwritten documents with multimodal LLMs. Two different prompting strategies are proposed to evaluate the impact of giving to an MLLM the OCR text for the whole document plus one page image, considering that a single image captures cross-page regularities (writer style, OCR error patterns) and it is sufficient to correct OCR on all pages.

The paper also presents a multi-page benchmark from IAM and Bentham and releases a new historical dataset (Malvern-Hills). Across three datasets, they compare page-by-page vs. all-at-once pipelines and text-only vs. image-only vs. hybrid inputs using GPT-4o, Gemini-2.5-Pro, and Gemma-3-27B.

Results show that OCR+PAGE1/PAGEN are consistently on the cost/accuracy Pareto frontier, often outperforming full image inputs in character error rate (CER) while being cheaper in tokens. The authors also present an LLM-based semantic error report showing fewer major errors for the proposed methods.

**Strengths:**

Treating multi-page handwriting recognition as a cross-page context sharing problem and asking an MLLM to learn the OCR to image error mapping from a single page is an original and practical idea.

The demonstration that one page’s visual signal often suffices to correct systematic OCR errors across unseen pages is both novel and impactful for cost-constrained digitization workflows.

The release of a multi-page benchmark is valuable.

The broad comparison across methods (PBP vs. AAO; OCR, images, hybrids) and models (Gemini, GPT-4o, Gemma) on three datasets, with cost accounting and a semantic error audit is important.

**Weaknesses:**

Despite the interesting results, the study does not yet position itself against the strongest non-MLLM baselines (e.g., modern HTR models with few-shot adaptation or lexicon-aware decoders) nor against competitive open MLLMs specialized for document OCR. The authors should argue more in this direction in the paper.

I am not sure I understand correctly, but it seems to me that combining single pages by writer ID likely increases lexical overlap across pages, making cross-page extrapolation easier than in organically multi-page documents.

Costs sometimes favor OCR+PAGE methods despite extra OCR tokens. In this case, the per-stage token counts, image-token policy, and retry/failure rates are not tabulated.

All datasets are English/Latin scripts. I suggest an evaluation on diverse scripts or heavily structured forms or arguments in this direction in the paper.

The use of LLM-as-judge where one vendor model is used to assess outputs of competing vendor models is interesting, but it could be interesting to expand the auditor set (cross-vendor) and include limited human adjudication beyond 5 docs.

Costs appear to mix OCR and LLM usage. I think it is important to report per-stage token counts, image tokenization policy, truncation events, and failure/retry rates.

**Questions:**

Did you use OCR bounding boxes/layout?

Have you tried OCR text + token-level image crops only where the OCR is uncertain? This could lower costs further while isolating where images help.

How do accuracy and cost scale for 5–10 page documents?

---

> ### Author Response · Authors · 2025-11-25
> **Response to reviewer 71my (1/2)**
>
> We thank the reviewer for their detailed engagement with our work, and for their patience during the review process. We have run comprehensive additional experiments, following the suggestions of several reviewers, which we believe greatly improve on our existing work. We have summarised them in an official comment above as a response to all reviewers. We would be grateful if the reviewer would take into consideration these new results, which we refer to in this response.
>
> We attempt to address the reviewer’s concerns below.
>
> ---
>
> > Despite the interesting results, the study does not yet position itself against the strongest non-MLLM baselines (e.g., modern HTR models with few-shot adaptation or lexicon-aware decoders) nor against competitive open MLLMs specialized for document OCR. The authors should argue more in this direction in the paper.
>
> We thank the reviewer for this excellent suggestion. We have run additional experiments on `PyLaia`, in response to your suggestion of a lexicon-aware decoder, TrOCR (fine-tuned for handwritten text) as a second non-MLLM baseline, and `DocOwl2` as an open MLLM specialized for document OCR. Please see Tables R4 and R5 and accompanying discussion in the official comment.
>
> We note that our problem setting is zero-shot, to better reflect the circumstances faced most often by practitioners wishing to transcribe handwritten documents in the real world (i.e. no access to labeled data), so we do not fine-tune any of these models. As a result their performance on `Malvern-Hills-5+` (see above) is poor compared with commercial MLLMs and even the commercial OCR engine. `PyLaia` even performs poorly on `IAM`, on which it is trained. These results demonstrate that current truly open models and non-MLLMs do not appear to be able to compete with commercial tools at this challenging task without significant labelling and engineering effort. We will update the paper to include these results and argue this position more strongly.
>
> ---
>
> >  It seems to me that combining single pages by writer ID likely increases lexical overlap across pages, making cross-page extrapolation easier than in organically multi-page documents.
>
> This is an interesting point. In this paper we assume that a multi-page document possesses at least some degree of similarity between pages — this may be in handwriting, semantic content, image quality/style, document type or other. It could be argued that a document with no such similarities is not really a single document at all, but a collection of separate ones. Indeed, the types of multi-page documents one is likely to encounter in the real world, such as letters, notebooks, minutes, legal documents, etc., are likely to possess several such similarities across pages.
>
> With this assumption in mind, we believe our construction of `IAM` is a reasonable one, and mirrors our real-world datasets. Still, the point is interesting — how do our methods perform in the case of truly unrelated pages? We include an ablation on such a dataset, `IAM-5-Random`, in an official comment — please see Table R3 and the accompanying discussion. For these 5 page documents with 5 different authors and unrelated content, we see, as expected, a decrease in performance compared to `IAM-5`, but a single page is still able to provide some benefit.
>
> ---
>
> > Costs sometimes favor OCR+PAGE methods despite extra OCR tokens. In this case, the per-stage token counts, image-token policy, and retry/failure rates are not tabulated
> > ...
> > Costs appear to mix OCR and LLM usage. I think it is important to report per-stage token counts, image tokenization policy, truncation events, and failure/retry rates.
>
> We thank the reviewer for pointing this out. We have updated the paper with comprehensive breakdowns of tokens (input text, input image and output text) and costs (OCR, input, output) for experiments, and included failure rates and types per method.
>
> Tokenization policy for OpenAI models is based on the [`tiktoken`](https://github.com/openai/tiktoken) library. For `gemini` and `gemma`, text token counts are computed using the [`count_tokens` method](https://ai.google.dev/gemini-api/docs/tokens?lang=python#text-tokens) packaged with the Gemini API. Unfortunately this method [does not work correctly for image tokens](https://github.com/googleapis/python-genai/issues/1349), so we compute image tokens with a custom script that implements [this explanation](https://ai.google.dev/gemini-api/docs/tokens?lang=python#multimodal-tokens) of image tokenization from the Gemini API documentation.

---

> > ### Author Response · Authors · 2025-11-25
> > **Response to reviewer 71my (2/2)**
> >
> > > All datasets are English/Latin scripts. I suggest an evaluation on diverse scripts or heavily structured forms or arguments in this direction in the paper.
> >
> > We thank the reviewer for this excellent suggestion. We have added experiments on a new dataset, `CASIA-5`, consisting of 5-page documents handwritten in Chinese, and detail these results in an official comment above, and in the updated paper; please see Table R1 and the accompanying discussion. To summarize, we see that`OCR+PAGE-N` and `OCR+IMAGES` provide a similar performance boost over OCR alone, which supports our findings that a single well-chosen image can provide most of the benefit provided by the full set of images, and at a lower token/overall cost.
> >
> > ---
> >
> >
> > > The use of LLM-as-judge where one vendor model is used to assess outputs of competing vendor models is interesting, but it could be interesting to expand the auditor set (cross-vendor) and include limited human adjudication beyond 5 docs.
> >
> > We thank the reviewer for this suggestion. Due to the significant time required to perform the auditing described, we have not been able to do it so far during the rebuttal period, as we have focused on additional experiments. We will endeavour to provide a more thorough adjudication during the review period.
> >
> > ---
> >
> > > Did you use OCR bounding boxes/layout?
> > > Have you tried OCR text + token-level image crops only where the OCR is uncertain? This could lower costs further while isolating where images help.
> >
> > We did not use bounding boxes in our methods, nor have we experimented with selective image cropping of uncertain page sections. This is an interesting and promising idea, but is beyond the scope of this review period. We thank the reviewer for these suggestions and reserve them for future work.
> >
> > ---
> >
> > > How do accuracy and cost scale for 5–10 page documents?
> >
> > We thank the reviewer for this question. We have run additional experiments on one new dataset (`CASIA-5`) and three re-workings of existing datasets (`Malvern-Hills-5+`, `Malvern-Hills-10+` `IAM-5+`) to explore performance on datasets of documents with 5-13 pages. We kindly ask the reviewer to see our official comment and the updated Appendix in the paper for full details, but to summarise, in all cases we find that either the best-performing model is one of our single-page methods, or that single page methods perform similarly well to full `OCR+IMAGES`, at lower cost and with faster inference time. Furthermore, as our methods use only a single image regardless of document length, the cost savings only increase with longer documents.
> >
> > ---
> >
> > We thank the reviewer again for their meaningful engagement with our work, detailed feedback, and patience during the review process. We hope we have addressed the reviewer's concerns, and, in light of our response and comprehensive additional experiments, we ask the reviewer to kindly consider raising their score.
> >
> > We are happy to answer any further questions.

---

### Author Response · Authors · 2025-11-25
**Response to all reviewers — additional experimental results (1/5)**

# Summary of additional results

We are grateful to all reviewers for their detailed engagement with our work. We address individual reviewer concerns in replies, but we would like to draw all reviewers' attention to additional experiments which strengthen our paper and address multiple concerns. We thank the reviewers for their patience while we obtained these results, and apologize for the delay in this response.

For clarity, all new experiments and related discussion have been included in a new **Appendix E**, and the uploaded paper has been updated. In the final version these details will be incorporated into the main paper or existing/new Appendix sections as appropriate.

We reproduce discussion and experimental details below, and **use abridged tables** for the sake of brevity; full tables can be found in the updated paper.

---
### Results on document datasets with longer page counts (5-13 pages)
- We thank reviewers **71my**, **5mTv** and **sUZQ** for suggesting this.
- We re-work `IAM` and `Malvern-Hills` to generate three longer-document benchmarks: `IAM-5`, with 5 pages per document, `Malvern-Hills-5+`, with an average of  5.8 pages per document, and `Malvern-Hills-10+`, with an average of 11.5 pages per document.
- We also introduce a new Chinese handwritten benchmark with 5 page documents; see below.
- See Tables R1, R2, R3, R5 & R6 for new results on these benchmarks.

---
### Results on an additional, non-Latin dataset
- We thank reviewer **71my** for this suggestion.
- We introduce **`CASIA-5`**, a dataset of generated 5 page documents derived from CASIA HDB2.1Test; a **Chinese-language** handwritten dataset [1]
- See Table R1 for new results on this benchmark.

---
### Results on three additional methods that are not commercial MLLMs
- We thank reviewers **71my** and **tXKd** for this suggestion
-  [**`PyLaia`**](https://gitlab.teklia.com/atr/pylaia) \[2\]: a CNN-RNN-CTC recognizer trained on IAM-DB with lexicon-aware decoding and an n-gram language model. We evaluate PyLaia on IAM.
- **`TrOCR`** \[3\] (specifically [trocr-large-handwritten](https://huggingface.co/microsoft/trocr-large-handwritten)): a pre-trained Transformer model fine-tuned on handwritten text. We evaluate TrOCR on Malvern-Hills-5+
- [**`DocOwl2`**](https://huggingface.co/mPLUG/DocOwl2) \[4\]: an open MLLM specialized for document processing, which we evaluate on Malvern-Hills-5+
- See Tables R4 and R5 for these results

---
### Results on a new method, **`OCR+PAGE-R`**
- We thank reviewer **sUZQ** for this suggestion.
- `OCR+PAGE-R` is a middle-ground method between `+PAGE-1` and `+PAGE-N` which provides a *random* page image to the post-processing MLLM, rather than exclusively the first page, or a page chosen by another LLM.
- See Tables R1, R2, R3, R5 & R6 for new results with this method.

---
### Misc

- An ablation on a randomized 5-page version of the IAM dataset; **`IAM-5-Random`**, to demonstrate the case when multi-page documents share neither semantic content or a common author. See Table R3 for these results.
	- We thank reviewer **tXKd**, whose point about documents varying in writer identity and content prompted this.
- More detailed data from experiments (we thank reviewers **71my** and **sUZQ** for these suggestions):
	- Token breakdowns per method by image, input text and output text, and cost breakdowns per method by OCR, MLLM input and MLLM output
		- See Tables 12, 18 and 19 in Appendix E of the updated paper for these breakdowns for the original experiments in Section 5.
	- Inference times
	- MLLM failure rates
- Additional metadata for Malvern-Hills, including occurrence of various OCR challenges (pages with multiple writers, tables, distractor text, etc.), breakdown of images by author IDs, and counts of different page types; see Tables R7, R8 and R9.
- Using this metadata, we provide a comparison of performance on documents of specific types and with specific features for `Malvern-Hills-5+`; see Tables R10-12 in our response to reviewer tXKd, and Tables 15-25 in the paper.
  - We thank reviewer **tXKd** for the suggestion to provide this metadata and further analysis

---

[1] Liu, Cheng-Lin, et al. "CASIA online and offline Chinese handwriting databases." _2011 international conference on document analysis and recognition_. IEEE, 2011.

[2] Tarride, Solène, et al. "Improving automatic text recognition with language models in the pylaia open-source library." _International Conference on Document Analysis and Recognition_. Cham: Springer Nature Switzerland, 2024.

[3] Li, Minghao, et al. "Trocr: Transformer-based optical character recognition with pre-trained models." _Proceedings of the AAAI conference on artificial intelligence_. Vol. 37. No. 11. 2023.

[4] Hu, Anwen, et al. "mplug-docowl2: High-resolution compressing for ocr-free multi-page document understanding." _Proceedings of the 63rd Annual Meeting of the Association for Computational Linguistics (Volume 1: Long Papers)_. 2025.

---

> ### Author Response · Authors · 2025-11-25
> **Response to all reviewers — additional experimental results (2/5)**
>
> # Selected results and discussion
>
> ---
> ## Table R1: `CASIA-5`; Chinese handwritten dataset
>
> See Table 9 in Appendix E of the updated paper for the full table.
>
> ***Details.*** We take the CASIA-HWDB2.1 test split, which consists of 300 images of multiple lines of handwritten Chinese text. Similar to the process for IAM, we generate multi-page dataset `CASIA-5` by randomly grouping pages with the same author ID, such that handwriting, but not semantic content, is consistent. We use a subset of 30 documents of 5 pages each in our experiments. We use Google Cloud Vision as our OCR engine.
>
> | MLLM     | Method           | CER (%)  | Total Cost ($/1k docs) |
> | -------- | ---------------- | -------- | ---------------------- |
> | —        | Google OCR only  | 12.84    | 7.50                   |
> | `gpt-4o` | `IMAGES-AAO`     | 43.64    | 29.30                  |
> |          | `IMAGES`         | 42.40    | 30.81                  |
> |          | `OCR`            | 18.32    | 26.75                  |
> |          | **`OCR+PAGE-1`** | 14.48    | 28.31                  |
> |          | `OCR+IMAGES`     | 14.16    | 41.58                  |
> |          | **`OCR+PAGE-R`** | 13.91    | 28.41                  |
> |          | **`OCR+PAGE-N`** | 12.87    | 28.68                  |
> | `gemini` | `OCR`            | 33.17    | 24.87                  |
> |          | `IMAGES-AAO`     | 23.07    | 18.09                  |
> |          | `IMAGES`         | 13.55    | 19.08                  |
> |          | **`OCR+PAGE-1`** | 12.61    | 23.25                  |
> |          | **`OCR+PAGE-R`** | 10.13    | 23.19                  |
> |          | **`OCR+PAGE-N`** | *8.94*   | 23.42                  |
> |          | `OCR+IMAGES`     | **8.39** | 27.80                  |
>
> ### Discussion
> For this challenging Chinese handwriting dataset, we see that the best-performing model overall is `OCR+IMAGES`, i.e. a model that has access to both OCR prediction and the full set of document images. The OCR engine on its own is already quite effective, indeed, no method involving GPT-4o improves on it, and many significantly worsen it. The only methods that improve on OCR alone are those that use both OCR and at least one image as input to `gemini`. The fact that `OCR+PAGE-N` and `OCR+IMAGES` provide a similar performance boost over OCR alone supports our findings that a single well-chosen image can provide most of the benefit provided by the full set of images, and at a lower token/overall cost.
>
> ---
> ## Table R2: `IAM-5`
>
> See Table 10 in Appendix E of the updated paper for the full table.
>
> ***Details.*** We generate 5-page documents by the same method we generated `IAM`; by grouping random individual pages by author ID so handwriting (but not necessarily semantic content) is consistent. We use 100 page images to generate 20 5-page documents.
>
> | MLLM     | Method           | CER (%)  | Cost ($/1k docs) | Time (s/doc) |
> | -------- | ---------------- | -------- | ---------------- | ------------ |
> | —        | `OCR`            | 3.36     | 5.00             | —            |
> | `gemini` | `OCR`            | 1.37     | 11.51            | 11.3         |
> |          | `IMAGES-AAO`     | 0.54     | 11.18            | 7.2          |
> |          | `IMAGES`         | *0.52*   | 11.72            | 21.4         |
> |          | **`OCR+PAGE-R`** | *0.52*   | 12.48            | 8.3          |
> |          | **`OCR+PAGE-N`** | *0.51*   | 12.59            | 13.5         |
> |          | `OCR+IMAGES`     | **0.47** | 17.39            | 18.1         |
> |          | **`OCR+PAGE-1`** | **0.47** | 12.40            | 7.9          |
>
> ### Discussion
> We see similar results for the longer document case of `IAM-5` as we did for `IAM`. `OCR+PAGE-1` remains the (joint) top-performing model and all OCR + single-image methods perform approximately as well as the full `OCR+IMAGES` method, *despite lacking access to 80% of images* (and the comparatively poor performance of OCR alone), and correspondingly have lower costs and inference times.

---

> > ### Author Response · Authors · 2025-11-25
> > **Response to all reviewers — additional experimental results (3/5)**
> >
> > ## Table R3: `IAM-5-Random`; inconsistent handwriting and semantic content ablation
> >
> > See Table 11 in Appendix E of the updated paper for the full table.
> >
> > ***Details.***  Dataset generated in the same way as `IAM-5`, but we ensure that all 20 generated documents contain pages from 5 different authors.
> >
> > | MLLM     | Method           | CER (%)  |
> > | -------- | ---------------- | -------- |
> > | —        | `OCR`            | 3.78     |
> > | `gemini` | `OCR`            | 1.56     |
> > |          | **`OCR+PAGE-1`** | 1.31     |
> > |          | **`OCR+PAGE-R`** | 0.80     |
> > |          | `IMAGES`         | 0.76     |
> > |          | `IMAGES-AAO`     | 0.75     |
> > |          | **`OCR+PAGE-N`** | *0.70*   |
> > |          | `OCR+IMAGES`     | **0.64** |
> >
> > ### Discussion
> > As expected, the benefit of OCR+single-image methods is less pronounced when there is neither consistent handwriting nor semantic content across pages — *but there is still some benefit*. Though `OCR+PAGE-1` underperforms, `+R` and `+N` are perform comparably with methods that include access to all images, at lower cost and faster inference time. These results demonstrate:
> > - That our intuition about the benefit of `+PAGE-X` methods is likely correct; the more similar pages in a document are (in content, writing, etc) the more performance gain can be achieved with only a single page (and vice versa)
> > - That even in cases where pages are quite different, a single page can still provide performance benefit. This may be a result of other page similarities (e.g. image quality, document type), or an example of multi-modal inputs assisting with reasoning in general as a version of in-context learning
> >
> > ---
> > ## Table R4: `IAM` with `PyLaia`
> >
> > See Table 12 in Appendix E of the updated paper for the full table.
> >
> > ***Details.*** As our problem setting is zero-shot and document-level, i.e. no training/fine-tuning data, `PyLaia` (and `TrOCR`) poses a problem as it is (i) largely dependent on fine-tuning to achieve reasonable performance on out-of-sample data, and (ii) operates on the line level, rather than the page or document level.
> >
> > We use PyLaia out-of-the-box with the public `Teklia/pylaia-iam` model, a CNN–BLSTM–CTC recognizer trained on IAM. We enable `PyLaia`’s lexicon-aware decoding and use the bundled files from the model repository (`tokens.txt` `lexicon.txt`, `language_model.arpa.gz`). The language model is a 6-gram character LM trained on IAM.
> >
> > | Method                  | CER (%) | Cost ($/1k docs) |
> > | ----------------------- | ------- | ---------------- |
> > | `PyLaia`                | 6.51    | 0.00             |
> > | Azure OCR only          | 3.81    | 2.26             |
> > | `OCR+PAGE-N` → `gemini` | 0.63    | 6.71             |
> > ### Discussion
> > Despite being fine-tuned on IAM, `PyLaia` performs worse than any MLLM-based method, and worse than the Azure OCR engine. The second and third rows are reproduced from Table 1 in the paper, for reference.
> >
> > Additional experiments on `Malvern-Hills` with `PyLaia` resulted in close to 100% CER and so are not reported. Overall, it appears that `PyLaia` generalizes extremely poorly without additional engineering. We must conclude that MLLM- and Transformer-based methods are clearly dominant for HTR.

---

> > > ### Author Response · Authors · 2025-11-25
> > > **Response to all reviewers — additional experimental results (4/5)**
> > >
> > > ## Table R5:`Malvern-Hills-5+` with `TrOCR` and `DocOwl2`
> > >
> > > See Table 13 in Appendix E of the updated paper for the full table.
> > >
> > > ***Details.***
> > > - **`Malvern-Hills-5+`**: we group consecutive pages (i.e. pages that are continuous sections from the same book of minutes) from Malvern-Hills to produce a dataset of 24 documents from 140 images, each with a minimum of 5 pages: 11 documents have 5 pages, 9 have 6, 1 has 7 and 3 have 8 (average: 5.83 pages/doc)
> > > - **`TrOCR`:**
> > > 	- Since our setting is zero-shot, we do not additionally fine-tune `TrOCR` on our datasets, but we do use the [`trocr-large-handwritten`](https://huggingface.co/microsoft/trocr-large-handwritten) model, which is already fine-tuned on handwriting from IAM.
> > > 	- As `TrOCR` is intended for line images, we use the kraken command line tool to generate line sub-images of `Malvern-Hills` pages and concatenate results.
> > > - **`DocOwl2`:**
> > > 	- We use the [`DocOwl2` model hosted on HuggingFace](https://huggingface.co/mPLUG/DocOwl2) without additional fine-tuning.
> > > 	- We use a temperature of zero and a sufficiently high output token limit to ensure no truncation events.
> > > 	- We used 5 rounds of prompt iteration for each mode, testing on a small sample of 5 documents for each.
> > >
> > > | MLLM     | Method               | CER (%)  | Cost ($/1k docs) | Time (s/doc) |
> > > | -------- | -------------------- | -------- | ---------------- | ------------ |
> > > | —        | `DocOwl2`            | 92.08    | 0.0              | 234          |
> > > |          | `TrOCR`              | 31.43    | 0.0              | 990          |
> > > |          | Azure OCR only       | 13.96    | 5.83             | 0.0          |
> > > | `gemini` | `OCR`                | 7.05     | 29.46            | 24.0         |
> > > |          | `IMAGES-ALL-AT-ONCE` | 6.14     | 27.12            | 19.2         |
> > > |          | **`OCR+PAGE-R`**     | 5.99     | 31.22            | 20.9         |
> > > |          | **`OCR+PAGE-N`**     | 5.86     | 31.43            | 31.2         |
> > > |          | `OCR+IMAGES`         | 5.69     | 35.75            | 27.5         |
> > > |          | `IMAGES`             | *5.63*   | 27.37            | 30.5         |
> > > |          | **`OCR+PAGE-1`**     | **5.43** | 31.15            | 21.4         |
> > >
> > > ### Discussion
> > > As with `IAM-5`, even with longer page counts, a `PAGE-X` method remains the best-performing, despite having access to <20% of the raw images per document, along with OCR which has a high error rate on its own. `IMAGES` alone performs quite well, and is slightly cheaper than `OCR+PAGE-1` due to the high text density of the `Malvern-Hills` dataset, but it is significantly slower as each image in a multi-page document requires a separate API call.
> > >
> > > Unfortunately, our `DocOwl2` and `TrOCR` non-commercial-MLLM baselines perform poorly on this task zero-shot, and are quite slow. `TrOCR`, at least, yields parsable text, but is prone to OCR-like errors such as the misreading of individual words or characters. Conversely, `DocOwl2` is occasionally accurate, but is extremely prone to a number of well-known LLM issues that destroy its overall performance: (i) early stopping (or simply outputting a single token), (ii) egregious hallucination often bearing no relation to the actual text, (iii) repetition of words or sentences ad infinitum, (iv) needless addition of code fences or explanatory text (e.g. double quotes, or 'the document reads...'), (v) repeating the prompt, and (vi) ignoring the prompt completely. These behaviours persisted regardless of explicit instructions given against them during prompt iteration.

---

> ### Author Response · Authors · 2025-11-25
> **Response to all reviewers — additional experimental results (5/5)**
>
> ## Table R6: `Malvern-Hills-10+`
>
> See Table 14 in Appendix E of the updated paper for the full table.
>
> ***Details.*** We use the same generation method as `Malvern-Hills-5+`. We obtain 10 documents in total from 115 images; 2 have 10 pages, 2 have 11, 5 have 12, 1 has 13 (average: 11.5 pages/doc).
>
> | MLLM     | Method           | CER (%)  | Cost ($/1k docs) | Time (s/doc) |
> | -------- | ---------------- | -------- | ---------------- | ------------ |
> | —        | `OCR`            | 12.92    | 11.50            | 0.0          |
> | `gemini` | `OCR`            | 6.88     | 59.29            | 47.9         |
> |          | `IMAGES-AAO`     | 6.05     | 54.46            | 40.9         |
> |          | **`OCR+PAGE-R`** | 5.50     | 61.82            | 40.1         |
> |          | **`OCR+PAGE-N`** | 5.41     | 61.77            | 63.8         |
> |          | `IMAGES`         | 4.87     | 54.71            | 60.6         |
> |          | `OCR+IMAGES`     | *4.80*   | 71.41            | 54.9         |
> |          | **`OCR+PAGE-1`** | **4.76** | 61.13            | 40.6         |
> ### Discussion
> Even for documents of at least 10 pages in length, `OCR+PAGE-1` remains the top-performing method, with `OCR+IMAGES` being the second best, about equivalent in performance, but much slower and more expensive. This suggests that using a single page to improve performance does scale reasonably well, even for quite long documents.
> This is good news, as it means that, for documents where the average number of text tokens inside each image is less than the average number of tokens produced by tokenizing the image (the case for `CASIA`, `Bentham`, `IAM` and many other real-world datasets — `Malvern-Hills` is particularly token-dense), then `+PAGE-X` methods will only be more cost and time effective in comparison to full-image methods as document length increases.
>
> ---
>
> ## Tables R7-9: `Malvern-Hills` metadata
>
> See Table 17 in Appendix E of the updated paper to see additional information detailing years in which images were written.
>
> **Table R7:** `Malvern-Hills` image statistics and occurrence of various OCR challenges.
>
> | **Statistic**                     |  **Value** |
> | --------------------------------- | ---------: |
> | Word count                        |   219 ± 83 |
> | Character count                   | 1260 ± 473 |
> | Includes tabular information      |      19.3% |
> | Includes margin notes             |      15.7% |
> | Includes distractor text          |      62.1% |
> | Non-linear structure              |       0.7% |
> | Archaic language                  |      31.4% |
> | Poor quality image/damaged paper  |       2.1% |
> | Handwriting from multiple authors |      18.6% |
> | Includes crossed-out text         |      23.6% |
>
> ---
>
> **Table R8**: Unique writers for the `Malvern Hills` dataset images and number of documents containing that writer's handwriting for each. Note that some pages contain multiple hands (see Table R7).
>
> |**Author ID**|**Count**|
> |--:|--:|
> |0|44|
> |1|23|
> |2|18|
> |3|17|
> |4|16|
> |5|15|
> |6|11|
> |7|6|
> |8|5|
> |9|5|
> |10|2|
> |11|2|
> |12|1|
> |13|1|
>
> ---
>
> **Table R9**: Breakdown of primary and secondary page types found in `Malvern-Hills`.
>
> | Primary Type                  | # Pages | Secondary Types               | # Pages |
> | ----------------------------- | ------: | ----------------------------- | ------: |
> | Historical minutes            |      98 | ---                           |      72 |
> |                               |         | Tabular                       |      26 |
> | Historical legal              |      34 | Statute                       |      17 |
> |                               |         | Memoranda roll                |      10 |
> |                               |         | Case memorandum               |       4 |
> |                               |         | Memoranda roll, Tabular       |       1 |
> |                               |         | Case memorandum, Legal letter |       1 |
> |                               |         | Legal letter                  |       1 |
> | Historical inventory/schedule |       8 | ---                           |       8 |

---

### Meta-Review · Area_Chair_7qa3 · 2026-01-06

**Summary:**

The submission received mixed evaluations, with two positive and two negative scores.

Reviewer 71my primarily calls for more extensive evaluations, including comparisons with non-MLLM baselines, experiments on diverse scripts, and a clearer cost analysis. Reviewer 5mTv, who assigned the lowest score (2 – reject), questions the novelty of the work (echoing Reviewer tXKd) and, like Reviewer sUZQ, criticizes the evaluation for relying on generally short multi-page documents.

**Reviewer Concerns:**

The authors have provided a detailed, point-by-point response to the reviewers’ comments. Upon review, the AC finds that most concerns have been adequately addressed, but believes the paper would benefit from clearer positioning. The limited technical novelty noted by reviewers is not considered a critical weakness. The AC agrees that the primary contribution is a thorough analysis of handwriting text recognition; however, as suggested by reviewers, the evaluation should be expanded to include more diverse and challenging scenarios, along with broader comparisons to related methods. If the introduction of the Malvern-Hills dataset is intended as a significant contribution, the AC would also expect a more in-depth analysis of this dataset—similar to most dataset papers—to better reveal insights.

**Reviewer Scores:**

All the scores will not be changed.

---

### Decision · Program_Chairs · 2026-01-26

Reject